# Balancing Model Performance and Rapid Personalization in Federated Learning with Learning Rate Scheduling

## Abstract

Federated learning (FL) is a powerful technique for collaboratively training a single centralized model on distributed local data sources. By aggregating model information without disclosing the local training data, FL preserves data privacy. However, the inherent heterogeneity in local data sets challenges the performance of FL techniques, especially when data is very diverse across local sources. Personalized Federated Learning (PFL) can mitigate this challenge using multiple models but often requires additional memory and computation. This work does not propose a new PFL method but introduces how learning rate decay, within each local training round, can improve model performance across local data sets after fine-tuning. We provide theoretical insights and empirical evidence of efficacy across diverse domains, including vision, text, and graph data. Our extensive experiments demonstrate that learning rate scheduling alone outperforms other FL methods regarding generalization to new data for both new and existing users. Moreover, it performs comparably to PFL methods, particularly regarding new users, while maintaining similar computation and memory requirements as FL techniques.

## 1 Introduction

Federated Learning (FL) is a compelling solution for constructing a shared (global) model from several local data sources that inherently cannot be exchanged or aggregated (Yang et al., 2019; Kairouz et al., 2021; Li et al., 2020b; Mahlool & Abed, 2022; Zhang et al., 2021a; Dupuy et al., 2022). Learning without sharing data becomes particularly crucial when data privacy or security is paramount, as exemplified by healthcare applications (Li et al., 2020a; Rieke et al., 2020; Wang et al., 2023). FL operates through an iterative procedure involving rounds of model improvement. These rounds start with distributing the current global model to local entities (users) and selecting participants to contribute to the model update. The chosen users participate by training their local copies of the model using their respective data and returning the updated model. Next, aggregation of the returned models produces a new global model. The described process represents the typical procedure for FL solutions, which learns only one shared model for all users. The most widely embraced FL method is FedAvg (McMahan et al., 2017), which computes the new global model as the parameter-wise average of returned local models.

When users have similar data, a single model can perform well. However, when local data are very diverse, ensuring good performance across users with a single shared global model can be challenging (Qu et al., 2022; Caldarola et al., 2022). Recent works (Chen et al., 2022; Tan et al., 2022) argue that alternatively, the focus should be on Personalized Federated Learning (PFL) methods that output models tailored to the local data. However, learning additional models can require extra memory and computation, and managing multiple models in PFL may lead to heightened communication requirements between users, a bottleneck in real-world applications. We desire a method that can perform well with heterogeneous user data but with a similar computational cost to FL techniques.

Next, meta-learning has been successfully applied to PFL to address the challenge of diverse user data without requiring additional models during training. Instead of learning multiple models during the federated training process, meta-learning methods focus on producing a shared model that can quickly adapt to local

datasets through fine-tuning. This approach significantly reduces the computational and memory overhead associated with PFL methods. Building on the foundational work by Nichol et al. (2018), which demonstrated that meta-learning solutions inherently emphasize varying **fixed** balances between initial model success (shared model performance on local datasets) and rapid personalization (personalized model performance after fine-tuning), we propose an enhancement tailored for FL applications. While traditional meta-learning prioritizes adaptation across a wide distribution of tasks, the context of FL involves a more constrained task distribution: the users' local datasets. Due to this domain-specific constraint, focusing more on initial model success can lead to more prominent personalized model performance after fine-tuning. We leverage within-round learning rate decay to balance initial model performance and rapid personalization **flexibly**. By adjusting the decay hyperparameter, our method enables application-specific emphasis, providing a customizable approach to optimize performance for varying levels of data heterogeneity. Crucially, this modification generalizes the widely used FedAvg algorithm (McMahan et al., 2017), maintaining its computational simplicity and convergence rate (Li et al., 2020c), requiring no additional gradients or models to be computed or stored. In Section 3.2, we mathematically justify how this flexible balance leads to better performance across heterogeneous data scenarios, enabling improved outcomes for new and existing users.

Finally, in Section 4, we provide extensive experimental results spanning diverse data sets across vision, text, and graph applications. We show that within-round learning rate decay enhances FL techniques and closes remaining performance gaps with PFL methods without the additional computation and communication overhead required by such methods. Importantly, we consistently observe a 1 to 4 percentage point improvement in average test set accuracy for new and existing users over FedAvg.

The organization of this work is as follows. First, Section 2 briefly summarizes related work from personalized federated learning and meta-learning. Second, Section 3 introduces our approach, mathematically justifies its improved flexibility, and establishes its convergence rate. Finally, Section 4 gives empirical evidence for performance improvements and computational cost compared with other benchmark methods.

## 2 Related Work

Federated Learning (FL) enables collaborative model training across distributed data sources without sharing raw data, preserving user privacy (Yang et al., 2019; Kairouz et al., 2021; Li et al., 2020b). However, data heterogeneity among users remains a critical challenge, leading to suboptimal global model performance for individual users. Various approaches have been proposed to address this, ranging from fine-tuning and meta-learning adaptations to fully personalized federated learning (PFL) techniques.

A simple yet effective strategy for personalization is fine-tuning, where the global model is adjusted using local user data post-training (Cheng et al., 2021; Chen et al., 2022; Tan et al., 2022). It is necessary to clarify that we consider fine-tuning separate from PFL solutions, as the additional computation and models happen after the federated training is completed. While computationally efficient during federated training, fine-tuning assumes access to sufficient local data, and its success largely depends on the quality of the shared initialization. Multiple works have found that FedAvg with fine-tuning typically outperforms more recent meta-learning and PFL methods (Cheng et al., 2021; Matsuda et al., 2023; Chen & Chao, 2021).

Meta-learning methods extend this idea by learning a globally shared model well-suited for rapid personalization (Jiang et al., 2019; Chen et al., 2018; Fallah et al., 2020a; Acar et al., 2021). In the context of FL, FOMAML (Jiang et al., 2019) and Per-FedAvg (Fallah et al., 2020b) have been adapted to optimize a shared initialization that accelerates convergence during fine-tuning. In addition to applying conventional meta-learning algorithms to the federated learning context, a variant named Reptile (Nichol et al., 2018) can be shown to be equivalent to FedAvg under the condition of equally sized local data sets. In this article, we exclusively consider aggregation with equally weighted users. Therefore, we will use the term FedAvg to encompass both FedAvg and Reptile and refer to FedAvg as a meta-learning method. The limitations of meta-learning approaches are that many require additional computation for second-order gradients, and the initialization often requires fine-tuning to perform well. More recent adaptations of meta-learning for FL include FedABML (Liu et al., 2023). However, in their experiments, FedABML typically performs around a single percentage point better than FedAvg with fine-tuning.

Next, PFL approaches address data diversity by learning personalized models for each user during federated training (Tan et al., 2023; 2022). Other personalization techniques than meta-learning include clustering (Mansour et al., 2020; Sattler et al., 2019; Briggs et al., 2020), model mixture (Li et al., 2021a; Mansour et al., 2020; Marfoq et al., 2021; Zhang et al., 2021b), parameter decoupling (Li et al., 2021b; Arivazhagan et al., 2019; Collins et al., 2023; Liang et al., 2020), and knowledge distrillation (Lin et al., 2020; Matsuda et al., 2021; Khalil et al., 2024; Shen et al., 2020). These intricate solutions often include learning a distinct model for each user throughout training (Li et al., 2021a), sharing only a subset of the entire model globally (Li et al., 2021b), or treating user data as a mixture of distributions (Marfoq et al., 2021). In general, PFL methods often require substantial increases in memory and computation to produce and communicate additional models during federated training, which challenges scalability and resource efficiency. Furthermore, several prevailing methods may prove restrictive or inequitable for applications accommodating new users. For instance, Ditto's approach (Li et al., 2021a) of training personalized models during federated learning becomes infeasible for users who do not participate in training. Likewise, FedBN (Li et al., 2021b) and FedEM (Marfoq et al., 2021), which rely on insights from user data, encounter difficulties when new users lack sufficient data for tailoring a model to new users.

Importantly, this work focuses on how including within-round learning rate decay can improve FL performance without significant extra computation or the need for additional models. Moreover, unlike the aforementioned meta-learning applications to FL, the learning rate decay can flexibly emphasize initial model performance and rapid personalization. We demonstrate that within-round learning rate decay is a generalization of FedAvg or Reptile, making them particularly relevant baselines for comparison. Several other works (Zhao et al., 2018; ?; Karimireddy et al., 2021; Reddi et al., 2021) propose to improve the original FedAvg procedure. However, many of these works improve the global optimization step. Unlike our work, they do not consider the local learning rate, especially within the communication round. While we do not propose a PFL method by definition, since fine-tuning occurs after federated training, we include PFL methods in our comparisons to assess the performance gap between our process and PFL techniques. There do exist more recent PFL solutions, like Marfoq et al. (2022), but comparison with PFL solutions is not the main focus of our work. More importantly, experiments demonstrate consistent improvements compared to FedAvg and FOMAML and close the performance gap with PFL methods with a simple, interpretable learning rate modification which comes a without additional computational or memory overheads.

## 3 Methodology

We consider Federated Learning constructed over $n = 1, \ldots, N$ communication rounds, each consisting of $k = 1, \ldots, K$ local update steps. The set $C = [1, \ldots, M]$ represents the users participating in federated learning, each with local objective function $F_i$. Federated learning aims to find the model that minimizes the average user objective. Note we distinguish $F_i$, which is evaluated on the entire local data set, from $F_i^{(n,k)}$, which is evaluated on data from the $k$-th local update step in round $n$. In each round, a subset of users, $S \subseteq C$, participates in the update of the global model, and the case $S = C$ is full user participation. We denote the global model at communication round $n$ as $\theta_g^{(n)}$, and $\theta_i^{(n,k)}$ represents the $i$-th user's copy of the shared model after $k$ of $K$ planned local update steps with learning rate $\eta$. This section begins with introducing our notation for the federated learning process.

Before introducing the proposed method, we provide background information on the similarities and differences between the popular existing methods: FedSGD, FedAvg (McMahan et al., 2017), and FOMAML (Nichol et al., 2018). We believe this viewpoint of existing algorithms is essential for understanding the derivation and benefits of our proposal. Importantly, we justify that our proposed method can flexibly balance the goals of **initial model success**, shared model performance on the local datasets, and **rapid personalization**, personalized model performance on local data after fine-tuning. Mathematically, we define initial model success in Equation 1 as the average user objective with the shared model at the end of federated training. Similarly, rapid personalization in Equation 2 is the average user objective with the personalized model, fine-tuned with the local data, after federated training is complete as indicated by the

superscript $(N + 1)$.

$$F_{IMS} = \frac{1}{M} \sum_{i=1}^{M} F_i(\theta_g^{(N)}) \tag{1}$$

$$F_{RP} = \frac{1}{M} \sum_{i=1}^{M} F_i(\theta_i^{(N+1)}) \tag{2}$$

We remark that either of these objectives could be weighted. However, in this work, we focus on an algorithm that performs well for all users and treats all users equally. Note that $F_{IMS}$ is the same objective as commonly minimized in traditional Federated Learning and $F_{RP}$ for Personalized Federated Learning (and meta-learning). We will demonstrate that controlling the application-specific focus between initial model success and rapid personalization empirically improves personalization and can extend to new users without sufficient data for fine-tuning. Counter to intuition, we find that placing additional emphasis on initial model success, relative to previous applications of meta-learning, can improve performance after rapid personalization. We believe that this is applicable to the majority of federated learning scenarios where some similarities exist between users. Furthermore, we demonstrate how to maintain the same rate of convergence as FedAvg under certain assumptions while preserving the interpretability of the model performance trade-off mentioned above.

### 3.1   Global and Local Updates for Federated Learning

Consider any given communication round. To simplify the notation, we omit the communication round $n$ from the notation. We define the size $K$ sequence of loss functions denoted as $\{F_i^{(j)}\}_{j=0}^{K-1}$, where $K$ is the number of local update steps. Recall that $F_i^{(k)}$ is the local objective function of the $i$-th user evaluated on data from the $k$-th local update step. Let $g_i^{(k)} = \nabla F_i^{(k)}\left(\theta_i^{(k)}\right)$ denote the gradient of user $i$ on the $k$-th local update step. Equation 3 defines the standard local update equation for the FedAvg ($K > 1$) or FedSGD ($K = 1$) procedure, which are produced by users performing stochastic gradient descent for $K$ updates.

$$\theta_i^{(k)} = \theta_i^{(k-1)} - \eta g_i^{(k-1)} \text{ for all } k = 1, \dots, K \tag{3}$$

After $K$-steps of local training, we aggregate the updated models $\theta_i^{(K)}$ or, equivalently, the changes made by local updates as in Equation 4. The new shared model becomes the averaged element-wise parameters from the locally trained user models.

$$\theta_g \leftarrow \frac{1}{|S|} \sum_{i \in S} \theta_i^{(K)} = \theta_g + \frac{1}{|S|} \sum_{i \in S} \underbrace{\left(\theta_i^{(K)} - \theta_g\right)}_{-\eta \times g_{Method}} \tag{4}$$

Our comparison of various methods will focus on understanding their expressions of the $\theta_i^{(K)} - \theta_g$ term given in Equation 4. This expression is the change that a user's locally training made to the shared model. For the FedAvg procedure, we can use Equation 3 recursively to write

$$\theta_i^{(K)} = \theta_i^{(0)} - \eta \sum_{j=0}^{K-1} g_i^{(j)} \tag{5}$$

Noting $\theta_i^{(0)} = \theta_g$, the change a user makes with the FedAvg procedure, $\theta_i^{(K)} - \theta_g$, is driven by $\sum_{j=0}^{K-1} g_i^j$. We refer to this component of a user's change as the **method gradient**, $g_{Method}$, due to its formulation and use in Equation 5. Similarly, other FL methods have a method gradient consisting of the gradients at local update steps. Equation 6 presents the specific gradients used from Equation 3 for FedSGD, FedAvg

(McMahan et al., 2017), and FOMAML (Nichol et al., 2018).

$$g_{FedSGD} = g_i^{(0)}, \quad g_{FedAvg} = \sum_{j=0}^{K-1} g_i^{(j)}, \quad g_{FOMAML} = g_i^{(K-1)} \tag{6}$$

Notably, the above methods result in local changes to the global model based on sums of different local gradients computed to update the model. In the next section, we demonstrate which and how much of each local gradient used during local training determines how much emphasis is placed between minimizing Equation 1 and Equation 2. Furthermore, we discover that within-round learning rates can flexibly emphasize objectives of initial model success and rapid personalization based on the decay used. In Section 4, we show that choosing decay that places additional emphasis on Equation 1, relative to FedAvg and FOMAML, results in solutions with improved personalized performance, despite Equation 2 being the objective associated with rapid personalization.

### 3.2   FedDecay: Generalizing Local Updates With Gradient Decay

**Within-Round Learning Rate Decay.**   From Equation 6, FedSGD, FedAvg, and FOMAML fully include or omit gradient terms from the summation. We introduce within-round learning rate decay, **FedDecay**, to enable a more general use of the local gradients. Within-round local learning rate decay is a simple, intuitive modification that allows for flexible, application-specific control over how much local gradients are used in local training, which determines the emphasis between minimizing Equation 1 and Equation 2. Equation 7 gives exponential within-round learning rate decay, which leads to Equation 8, the method gradient for FedDecay. Our proposed method generalizes previous work by recovering FedSGD when $\beta = 0$ and FedAvg when $\beta = 1$.

$$\theta_i^{(k)} = \theta_i^{(k-1)} - \eta \beta^{k-1} g_i^{(k-1)} \text{ for all } k = 1, \dots, K \tag{7}$$

$$g_{FedDecay} = \sum_{j=0}^{K-1} \beta^j g_i^{(j)} \tag{8}$$

Note, a within-round local learning rate of the form $\beta^{k-1}$ in Equation 7 results in less emphasis on the successive local gradients for smaller $\beta$ in Equation 8. Importantly, we next identify that the later local gradients are associated with rapid personalization. Thus, by introducing within-round local learning rate decay, we can place more relative emphasis on initial model success.

This work primarily focuses on exponential decay, simplifying the mathematical justification for the benefits of within-round learning rate decay. Exponential decay allows us to explicitly compute the relative emphasis between initial model success (minimizing Equation 1) and rapid personalization (minimizing Equation 2). Our primary theoretical contribution is determining that the within-round learning rate controls the prioritization of the above objectives. In Appendix A.1, the coming results for FedDecay are presented more generally for arbitrary positive sequences. Additionally, Appendix B.2 demonstrates that linear decay can improve performance.

**Balancing Initial Model Success and Rapid Personalization.**   The justification for FedDecay comes from a Talyor analysis of the method gradients from Equations 6. As found by Nichol et al. (2018), each method gradient in Equations 6, in expectation, is approximately the weighted sum of two other gradient terms: one which promotes initial model success, and the second which prioritizes rapid personalization. Consider the $K$-length sequence of loss functions $\{F_i^{(j)}\}_{j=0}^K$ for the $i$-th user's local update. Let $\tilde{g}_i^{(j-1)} = \nabla F_i^{(j-1)}(\theta_g)$ and $\tilde{H}_i^{(j-1)} = \nabla^2 F_i^{(j-1)}(\theta_g)$ denote the gradient and Hessian, respectively, of the $j$-th loss function evaluated instead at the most recent global model. We take expectation over the user $i$ and minibatches $k$ and $l$ from the local data. Following the work of Nichol et al. (2018), we refer to those expected gradient terms $\mathbb{E}[\tilde{g}_i^{(k)}]$ and $\mathbb{E}[\tilde{H}_i^{(k)} \tilde{g}_i^{(l)}]$ as AvgGrad and AvgGradInner, respectively. First, moving in the negative direction of AvgGrad attempts to minimize the expected loss across users and data, encouraging the final shared model to perform well on all user data, otherwise known as initial model success. Second, moving

in the positive direction of AvgGradInner maximizes the inner product between gradients (as illustrated in Equation 9 by the chain rule) from distinct mini-batches within the same user. Large inner products indicate that the gradient will recommend moving in a similar direction regardless of the input data, which helps facilitate rapid personalization.

$$\mathbb{E}\left[\tilde{H}_i^k \tilde{g}_i^l\right] = \frac{1}{2}\mathbb{E}\left[\tilde{H}_i^k \tilde{g}_i^l + \tilde{H}_i^l \tilde{g}_i^k\right] = \frac{1}{2}\mathbb{E}\left[\frac{\partial}{\partial \theta_g}\left(\tilde{g}_i^k \tilde{g}_i^l\right)\right] \tag{9}$$

Next, we will show that incorporating within-round learning rate decay allows for a flexible emphasis on each objective, which can be optimized by tuning the hyperparameters associated with the given decay scheme. Equation 10 gives the expected method gradient of FedDecay for exponential decay and exists in greater generality in Section A.1.

$$\mathbb{E}\left[g_{FedDecay}\right] \approx \left(\frac{1-\beta^K}{1-\beta}\right)\mathbb{E}\left[\tilde{g}_i^{(k)}\right] - \left(\frac{(1-\beta^{K-1})(1-\beta^K)}{(1+\beta)(1-\beta)^2}\right)\mathbb{E}\left[\tilde{H}_i^{(k)} \tilde{g}_i^{(l)}\right] \tag{10}$$

Importantly, the inclusion of within-round learning rate decay results in coefficients for each gradient term of interest, that depend on the decay hyper-parameter $\beta$. More relative emphasis will be placed on AvgGrad when $\beta$ is small, and conversely AvgGradInner when $\beta$ is large.

As stated previously, AvgGrad and AvgGradInner also appear in the expected method gradients of FedSGD, FedAvg, and FOMAML (Nichol et al., 2018). By comparing the ratio of the weights of AvgGradInner to AvgGrad for the considered methods, we can assess each method's relative prioritization of rapid personalization vs. initial model success. See Table 1 and contrast the more flexible ratio of FedDecay with those of FedSGD, FedAvg, and FOMAML. Notably, FedSGD, FedAvg, and FOMAML have fixed ratios, given $K$. On the other hand, the ratio of FedDecay still depends on the choice of $\beta$ giving us much greater flexibility over the balance between initial model success and rapid personalization. Especially in applications where communication is a bottleneck, $K$ may be required to be large. In such applications, methods such as FedAvg and FOMAML, which place increasing emphasis on $K$, are forced to prioritize rapid personalization over initial model success, even if users all had copies of the exact same data. In Section **??**, we provide strong empirical evidence that, in general, placing additional emphasis on initial model success can result in improvements to performance after personalization.

| Method | FedSGD | **FedDecay** | FedAvg | FOMAML |
|--------|--------|--------------|--------|--------|
| Ratio | $\dfrac{0}{1}$ | $\dfrac{\left(\beta - \beta^K\right)\eta}{1-\beta^2}$ | $\dfrac{(K-1)\eta}{2}$ | $(K-1)\eta$ |
| Limit $(K \to \infty)$ | $0$ | $\dfrac{\beta\eta}{1-\beta^2}$ | $\infty$ | $\infty$ |

Table 1: Ratios of coefficients for AvgGradInner (rapid personalization) to AvgGrad (initial model success) terms for several methods. Choice of $\beta$ gives FedDecay much greater control over the ratio of emphasis on the ability to personalize vs. initial model success. Note that we are assuming that $\beta \in (0,1)$ for FedDecay since FedSGD ($\beta = 0$) and FedAvg ($\beta = 1$) are already present.

When users possess similar data, a method consisting of only AvgGrad terms (FedSGD) can perform well across all users. However, in cases where users' data diverges, an approach prioritizing AvgGradInner terms may be more suitable. Plausibly, most applications exist between the above two scenarios, especially for appropriate applications of federated learning. In such cases, a method striking a flexible balance between AvgGrad and AvgGradInner terms could yield superior performance across a spectrum of statistical data heterogeneity. Our experimental results in Section **??** consistently conclude that some decay improves personalized performance compared to either no emphasis on rapid personalization (FedSGD) or no decay (FedAvg). Furthermore, in Appendix B.1 we highlight that the tuned values of $\beta$ are small for datasets with obvious similarities between users and greater for datasets contrived to have a larger degree of diversity between local datasets.

In summary, introducing within-round learning rate decay enables more control over the objective of training. This additional flexibility allows FedDecay to adapt to the problem-specific data heterogeneity for

better performance. First, when local datasets are similar, a large amount of decay (small $\beta$) still places some emphasis on rapid personalization that results in improved personalized performance when fine-tuning. Secondly, when local datasets have some minor similarities, placing the majority of the focus on rapid personalization, but more emphasis on initial model success with a small amount of decay (large $\beta$) than FedAvg and FOMAML also improves personalized performance. We believe intuitively that having a shared model that performs well for all users and fine-tunes well is often better than a model that fine-tunes well, but may not have any initial model success, which is not a prerequisite for the rapid personalization objective in Equation 2.

**Convergence Analysis.** Within-round learning rate decay allows for an application-specific hyper-parameter tuning of the emphasis between initial model success and rapid personalization. Separate from the purpose of our work, Li et al. (2020c) demonstrate that the convergence of FedAvg on non-independent and non-identically distributed (i.i.d.) local data sets is achieved with some learning rate at iteration $t$ of the form $\eta_t = \alpha_t = \frac{c}{t+d}$ where $c$ and $d$ are positive. Note that the local update step $k$ of the communication round $n$ is equivalent to iteration $t = n * K + k$. We remark that this learning rate strategy is not within-round and does not exist for understanding data heterogeneity. In fact, we argue that this strategy is insufficient for convergence and balancing initial model success with rapid personalization. Performing federated training for many communication rounds is needed to guarantee a close-to-optimal objective function value. However, such decay of the learning rate every iteration places more relative emphasis on rapid personalization with each communication round. Unfortunately this scheme results in a greater relative emphasis on personalization with each addition communication round, similar to how FedAvg and FOMAML were found to place additional relative emphasis on personalization for many local update steps ($K$). Hence, we lose the ability to prioritize initial model success for users with similar data with decay of the learning rate every iteration. Additionally, we lose all interpretability about the degree of heterogeneity based on the size of the tuned values for $\beta$. We no longer are able to infer that applications that perform well with large $\beta$ have similar local datasets (or conversely small $\beta$.

Instead, by composing our within-round learning rate decay, with previous decay applied across-round (instead of each iteration), we can maintain the same convergence rate as FedAvg (Li et al., 2020c) while keeping the relative balance between initial model success and rapid personalization constant across communication rounds. The resulting learning rate is given by $\eta_t = \alpha_{\lfloor t/K \rfloor}\beta_{(t \bmod K)}$ where $\beta_0 \neq 0$ and $\beta_{k+1} \leq \beta_k$ for $k = 1, \ldots, K$. We extend the previous work Li et al. (2020c) to this modified decay scheme under the same assumptions. FedDecay converges with complexity of $\mathcal{O}(N^{-1})$ on non-i.i.d. data as total communication rounds $N \to \infty$ for both full and partial user participation. Please see Appendix A.2 for additional details and proofs. Note that we do not believe that the theoretical finding that applying the previous decay scheme across-round converges in particularly novel, nor do we consider it a primary contribution of our work. However, we believe that having explicit convergence guarantees for application of our proposal are essential to creating confidence in its application. Hence, we believe the important takeaway from this section is that our proposal allows us adapt to application-specific data heterogeneity by adjusting the relative prioritization between objectives of initial model success and rapid personalization without sacrificing theoretical convergence.

**Assumption 1** (Lipschitz Continuous Gradients)**.** *For all users $i \in \{1, \ldots, M\}$, the local objective function $F_i$ is $L$-smooth. For all $v$ and $w$ in the domain of $F_i$,*

$$F_i(v) \leq F_i(w) + (v - w)^T \nabla F_i(w) + \frac{L}{2} \|v - w\|_2^2$$

**Assumption 2** (Strong Convexity)**.** *For all users $i \in \{1, \ldots, M\}$, the local objective function $F_i$ is $\mu$-strongly convex. Similarly,*

$$F_i(v) \geq F_i(w) + (v - w)^T \nabla F_i(w) + \frac{\mu}{2} \|v - w\|_2^2$$

**Assumption 3** (Bounded Variance)**.** *Let mini-batch $\xi^{(n,k)}$ be sampled uniformly from the $i$-th user's local data. For all $n$ and $k$, an upper bound exists on the variance of stochastic gradients in each user $i$.*

$$\mathbb{E} \left\| \nabla F_i^{(n,k)} \left( \theta_i^{(n,k)}, \xi_i^{(n,k)} \right) - \nabla F_i \left( \theta_i^{(n,k)} \right) \right\|^2 \leq \sigma_i^2$$

where $F_i^{(n,k)}$ is the average for $i$-th user's local loss evaluated on data $\xi_i$.

**Assumption 4** (Bounded Expected Squared Norm). *The expected squared norm of stochastic gradients is uniformly bounded. Similarly,*

$$\mathbb{E}\left\|\nabla F_i^{(n,k)}(\theta_i^{(n,k)}, \xi_i^{(n,k)})\right\|^2 \le G^2$$

Here, we consider the more general weighted objective function $F = \sum_{i \in C} p_i F_i$ and aggregation step given by Equation 11. Recall that $F_i$ is the average local loss over all of the local data of the $i$-th user and $S \subseteq C$ of fixed size $|S|$ contains the users chosen to update the global model in a given round of Federated Learning. $S$ is sampled uniformly without replacement for each round with probabilities $p_i$.

$$\theta_g \leftarrow \frac{|C|}{|S|} \sum_{i \in S} p_i \theta_i^{(K)} \tag{11}$$

Furthermore, let $F^*$ and $F_i^*$ denote the global minimums of the average objective function of users and the $i$-th user's average local loss over their data, respectively. We introduce $\Gamma = F^* - \sum_{i=1}^M p_i F_i^*$ to represent the degree to which data sets are non-independent and non-identically distributed. The magnitude of $\Gamma$ reflects how heterogeneous the data distributions are and $\Gamma \to 0$ as the number of local data samples grows only when data are iid.

**Theorem 1** (Convergence Of FedDecay). *Let Assumptions 1 to 4 hold and $L$, $\mu$, and $G$ be defined therein and denote the condition number with $\kappa = L/\mu$. Choose a positive, locally decaying sequence $\beta_{k+1} \le \beta_k$ for $k = 1, \dots, K$. With a learning rate for iteration $t = nK + k$ ($k$ local updates into round $n + 1$) of the form $\eta_t = \alpha_{\lfloor t/K \rfloor} \beta_{t \bmod K}$ for positive $\beta_j \ge \beta_{j+1}$ and some $\alpha_j = \dfrac{c}{j+d}$. If $\alpha_j = \dfrac{3}{\mu \beta_{K-1}(j+d)}$ where $d = \max\left\{\dfrac{12\kappa}{\beta_{K-1}}, 4 - \dfrac{2}{K}\right\}$ then*

$$\mathbb{E}\left[F(\theta_g^{(N)})\right] - F^* \le \frac{\kappa}{N+d-2}\left(\frac{9(B+D)}{2\mu \beta_{K-1}^2} + \frac{(1/K)+d-2}{2} \times \mathbb{E}\left\|\theta_g^{(0)} - \theta_g^*\right\|^2\right) \tag{12}$$

$$\text{where } B = \sum_{i=1}^M p_i^2 \sigma_i^2 + 6L\Gamma + 2\left(G(K-1)\beta_0 \beta_{K-1}^{-1}\right)^2 \tag{13}$$

$$\text{and } D = \frac{|C| - |S|}{|S|\,(|C| - 1)} \tag{14}$$

The convergence analysis outlined in Theorem 1 emphasizes that the gap between the optimal solution $F^*$ and the value $F(\theta_g^{(N)})$ evaluated on the global model, produced by FedDecay after $N$ total communication rounds, converges to zero as $N \to \infty$. Akin to the outcomes seen with FedAvg, convergence happens at rate $\mathcal{O}(N^{-1})$. However, the advantage of FedDecay lies in its ability to achieve this convergence while accommodating a spectrum of diverse local updates from the sequence $\{\beta_k\}_{k=1}^K$. As illustrated in Section 3.2, incorporating within-round learning rate decay enables a versatile balance between initial model success and rapid personalization that allow us to adapt to application-specific data heterogeneity. In conclusion, this enriched flexibility does not have to come at the expense of deteriorating algorithmic performance.

## 4 Experimental Results

In this section, we comprehensively evaluate our proposed method, FedDecay, alongside other prominent federated learning techniques, using the benchmark established by Chen et al. (2022). To ensure uniformity, we integrate our approach into their code base available at GitHub [1] [2]. We leverage this established infrastructure to experiment with identical data sets and models to facilitate a rigorous and fair comparison.

---

[1]Original benchmark at `https://github.com/alibaba/FederatedScope/tree/Feature/pfl_bench`
[2]Our modifications at `https://github.com/JoeLavond/FedDecay/tree/bench`

**Data.** As discussed in Yuan et al. (2022), generalization in federated learning refers both to new users (participation gap) and new data for existing users (out-of-sample gap). Our experiments involve holding out both users and data to study both types of generality in the following manner. For each data set, we maintain 20 percent of users as a holdout set, thereby simulating the challenge of generalization to new users. Also, we divide each user's data into distinct training, validation, and testing subsets. Our experiments encompass the various data sets FEMNIST (Caldas et al., 2018), SST2 (Wang et al., 2018; Socher et al., 2013), and PUBMED (Namata et al., 2012). This diversified selection of data sets empowers us to investigate performance across varied domains, including vision, text, and graph data. We use the default data settings chosen by Chen et al. (2022).

- FEMNIST: A 62-way handwritten character classification problem with images of size $28 \times 28$. This sub-sampled version of FEMNIST in Chen et al. (2022) contains 43,400 images from 3,550 authors as clients.

- SST2: A sentiment classification data set containing 68,200 movie review sentences labeled with human sentiment. Partitioned into 50 clients using Dirichlet allocation with $\alpha = 0.4$, it enables the assessment of text-based applications.

- PUBMED: A graph of 19,717 nodes and 44,338 edges, this data set classifies scientific publications into three classes. Louvain (Blondel et al., 2008) community partitioning parititons the graph into five users.

**Model.** In our experiments, we adhere to the model architectures prescribed by Chen et al. (2022) for consistency and comparative purposes.

- FEMNIST: We employ a Convolutional Neural Network (CNN) with a hidden size of 2,048. The model incorporates two convolutional layers with $5 \times 5$ kernels, followed by max pooling, batch normalization, ReLU activation, and two dense layers.

- SST2: Our model leverages a pre-trained BERT-Tiny model (Turc et al., 2019) with 2-layer Transformer encoders and a hidden size 128.

- PUBMED: We employ the Graph Isomorphism Neural Network (GIN) (Xu et al., 2019), featuring 2-layer convolutions with batch normalization, a hidden size 64, and a dropout rate of 0.5.

**Methods.** To establish the prowess of FedDecay, we subject it to a rigorous comparison with a range of federated learning techniques.

- Federated Learning: In this category, we include FedAvg/Reptile (McMahan et al., 2017; Nichol et al., 2018) and FOMAML (Nichol et al., 2018). These methods make up the most relevant baselines to our proposal as they all can be seen, like our proposal, as various within-round learning rate scheduling.

- Personalized Federated Learning: These encompass Ditto (Li et al., 2021a), FedBN (Li et al., 2021b), FedEM (Marfoq et al., 2021), and pFedMe (T Dinh et al., 2020). Interpret these PFL solutions as an upper bound on what performance could be for our method as they require additional computation cost and memory.

**Hyperparameters.** We extend the hyper-parameter optimization methodology outlined in the original benchmark paper. Leveraging a grid search technique, we conduct comprehensive explorations, leveraging early termination and hyperband stopping (Biewald, 2020). Our exhaustive search covers a spectrum of hyperparameters, as detailed in Table 2.

The final hyperparameters for each run are selected based on the highest average validation accuracy across users. If runs do not terminate early, they will run for 1000, 500, and 500 epochs for FEMNIST, SST2, and PUBMED, respectively. Other fixed configurations worth noting are 20 percent partial user participation

| Hyper-parameter | Algorithm | Data Set | Tuning Grid |
|---|---|---|---|
| Local Epochs | - | - | $\{1, 3\}$ |
| Batch Size | - | SST2 | $\{16, 32, 64\}$ |
| Learning Rate | - | - | $\{0.005, 0.01, \ldots, 1.0\}$ |
| Regularization Rate | Ditto, pFedMe | - | $\{0.05, 0.1, 0.2, 0.5, 0.9\}$ |
| Meta-learning Step | pFedMe | - | $\{1, 3\}$ |
| Mixture Number | FedEM | - | $\{3\}$ |
| Local Decay | FedDecay | - | $\{0, 0.2, \ldots, 1.0\}$ |

Table 2: Hyper-parameter grid search details. If no algorithm or data set is specified, the given hyper-parameter search will be applied for all.

for FEMNIST and SST2, batch size 32 for FEMNIST, and full-user participation with full-batch training for PUBMED. To ensure a fair comparison and alignment with other methods, we maintain a fixed learning rate ($\alpha$) for FedDecay, while $\beta$ is the key hyperparameter explored. Fine-tuning happens for one epoch with the same learning rate $\alpha$ for all methods. We conduct our experiments on 4 NVIDIA GeForce RTX 3090 GPUs. See Appendix B.1 to analyze FedDecay's performance under misspecified $\beta$.

## 4.1 Generalization to New and Existing Users

**Generalization Performance On New Users.** To assess the robustness of the methods in accommodating new users, we introduce the following performance metrics: $\bar{Acc}$ (average accuracy), $\breve{Acc}$ (bottom ten percentile accuracy), and $\sigma_{Acc}$ (standard deviation of accuracy). Average accuracy is the primary performance metric, and it is desirable to identify methods that often return large values for $\bar{Acc}$, indicating that the solution performs well for users, on average. In addition, we use the fairness metrics $\breve{Acc}$ and $\sigma_{Acc}$ to understand if solutions are performing well for many users. A fair method consistently returns a large bottom ten percentile of accuracy and a low standard deviation, which indicates a lower bound on performance for the 90 percent majority of users and that the solution performs similarly across users. Please note that in the PUBMED data set comprising only five users, $\breve{Acc}$ and $\sigma_{Acc}$ are not applicable due to the limited number of held-out users. Additionally, we repeat the single-model-based results across multiple seeds in Appendix B.3 to gain insights into result variability and partially assess the sensitivity of our findings to different initial conditions.

We initiate our analysis by investigating the proficiency of each method in catering to new users. See Table 3 Our approach, FedDecay, notably outperforms all FL methods in terms of average test accuracy for all data sets. Furthermore, even in the FEMNIST data sets, it is worth highlighting that FedDecay significantly narrows the performance gap between FL and PFL methods. While personalized methods such as FedEM and pFedMe exhibit competitive results on FEMNIST, their performance drops noticeably in the SST2 data set. These findings underscore FedDecay's exceptional ability to generalize to new users across diverse applications effectively.

**Generalization Performance On New Data For Existing Users** Turning our attention to the performance of users who participate in the federated learning process, we anticipate that personalized federated learning methods, characterized by additional memory and computation for learning personalized models, should outperform FL. The results, presented in Table 4, affirm this expectation. However, among the FL methods, FedDecay emerges as the front-runner, achieving the highest average and bottom ten percentile test accuracy across all data sets. Furthermore, FedDecay even secures the highest average accuracy among all methods on the SST2 data set.

Delving into personalized federated learning techniques, we observe mixed results. Ditto and FedBN struggle on the SST2 data set, pFedMe faces challenges on PUBMED, and FedEM encounters difficulties on FEMNIST. Additionally, it is worth noting that FedDecay consistently outperforms other methods on at least one data set despite their additional computation and memory requirements. Please refer to Section 4.2 for a detailed exploration of the cost implications.

| Type | Method | FEMNIST (image) | | | SST2 (text) | | | PUBMED (graph) | | |
|---|---|---|---|---|---|---|---|---|---|---|
| | | $\bar{Acc}$ | $\breve{Acc}$ | $\sigma_{Acc}$ | $\bar{Acc}$ | $\breve{Acc}$ | $\sigma_{Acc}$ | $\bar{Acc}$ | $\breve{Acc}$ | $\sigma_{Acc}$ |
| Personalized Method | Ditto | 0.5672 | 0.4444 | 0.0921 | 0.4746 | 0.0000 | 0.4010 | 0.2442 | - | - |
| | FedBN | 0.9059 | 0.8302 | 0.0544 | **0.8030** | **0.6667** | 0.1275 | **0.8004** | - | - |
| | FedEM | **0.9175** | 0.8333 | 0.0526 | 0.7404 | 0.6379 | 0.2141 | 0.7879 | - | - |
| | pFedMe | 0.9036 | **0.8438** | 0.0751 | 0.7785 | 0.6406 | 0.1179 | 0.7932 | - | - |
| Single-model Based | FOMAML | 0.8989 | 0.8113 | 0.0604 | 0.7680 | 0.6667 | 0.1057 | 0.7950 | - | - |
| | FedAvg | 0.9055 | **0.8491** | 0.0530 | 0.7680 | 0.6667 | 0.1057 | 0.7914 | - | - |
| | **FedDecay** | **0.9152** | 0.8421 | 0.0485 | **0.8101** | **0.6724** | 0.1087 | **0.8039** | - | - |

Table 3: Generalization to new users who did not participate in federated training after fine-tuning. FedDecay has the most considerable average test set accuracy ($\bar{Acc}$) of any single-model-based technique on all data sets, even outperforming personalized methods on SST2 and PUBMED. Note that these PFL methods are upper bounds, not relevant baselines, due to their higher computational cost and memory requirements. With five total users for PUBMED, only a single user is held out to evaluate generalization. Hence, there are no values for the bottom ten percentile ($\breve{Acc}$) or standard deviation ($\sigma_{Acc}$) for new users.

| Type | Method | FEMNIST (image) | | | SST2 (text) | | | PUBMED (graph) | | |
|---|---|---|---|---|---|---|---|---|---|---|
| | | $\bar{Acc}$ | $\breve{Acc}$ | $\sigma_{Acc}$ | $\bar{Acc}$ | $\breve{Acc}$ | $\sigma_{Acc}$ | $\bar{Acc}$ | $\breve{Acc}$ | $\sigma_{Acc}$ |
| Personalized Method | Ditto | 0.9031 | 0.8333 | 0.0563 | 0.5949 | 0.0417 | 0.3449 | 0.8754 | 0.8465 | 0.0236 |
| | FedBN | 0.9182 | 0.8571 | 0.0548 | 0.7360 | 0.4000 | 0.2292 | 0.8788 | **0.8540** | 0.0248 |
| | FedEM | 0.8952 | 0.8200 | 0.0806 | **0.7808** | 0.5000 | 0.1845 | **0.8822** | 0.8478 | 0.0322 |
| | pFedMe | **0.9280** | **0.8750** | 0.0694 | 0.7699 | **0.6087** | 0.1672 | 0.8666 | 0.8205 | 0.0334 |
| Single-model Based | FOMAML | 0.8951 | 0.8140 | 0.0636 | 0.7654 | **0.5556** | 0.1634 | 0.8614 | 0.8292 | 0.0284 |
| | FedAvg | 0.8851 | 0.8039 | 0.0750 | 0.7654 | **0.5556** | 0.1634 | 0.8671 | 0.8342 | 0.0267 |
| | **FedDecay** | **0.8986** | **0.8214** | 0.0711 | **0.7815** | **0.5556** | 0.1727 | **0.8722** | **0.8490** | 0.0233 |

Table 4: Generalization to new data for existing users who participated in federated training after fine-tuning. FedDecay produces the best test set average and bottom ten percentile accuracy of any single-model-based method. Note that these PFL methods are upper bounds, not relevant baselines, due to their higher computational cost and memory requirements.

### 4.2 Computational and Memory Costs

The computational and communication costs incurred during training are integral to evaluating federated learning methods. Personalized federated learning methods, in particular, often demand increased computation and memory resources. In this context, we draw attention to the cost implications of various methods. As depicted in Figure 1, FedDecay exhibits comparable computational costs to other FL techniques. The cost of learning rate scheduling is negligible per iteration, and the potential increase in communication rounds due to decaying updates does not meaningfully affect computation. In our experiments, FedDecay returns an identical cost to FedAvg for FEMNIST and SST2, which terminate in the same number of epochs. FedDecay achieves similar or superior results without considerably higher communication and computation costs.

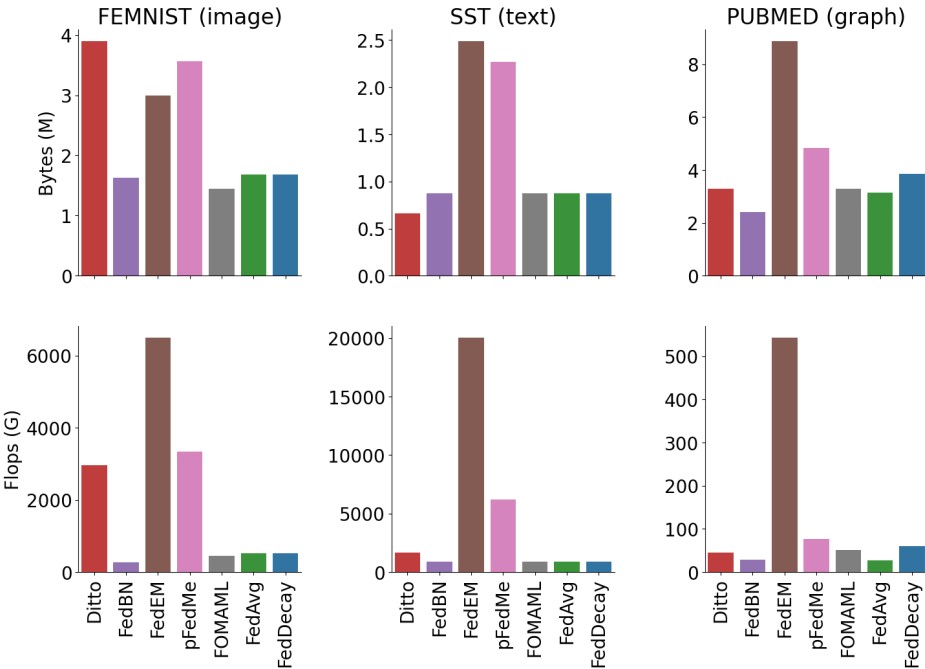

Figure 1: Computational And Memory Costs. Total bytes communication (*top*) and total floating point operations (*bottom*) performed by the best-performing run of each method. FedDecay requires similar costs to other FL techniques, like FedAvg. Furthermore, FedDecay incurs considerably lower costs than personalized methods, especially FedEM and pFedMe.

FedDecay requires substantially less memory and computation than FedEM and pFedMe, the two personalized methods that occasionally outperformed FedDecay in Section 4.1. Although FedDecay does not consistently outperform personalized federated learning methods, it is much more computationally efficient. It often dramatically closes the gap between other FL methods and personalized federated learning solutions. This realization emphasizes FedDecay's efficiency in striking a favorable balance between performance and resource utilization, making it an attractive solution for practical federated learning scenarios.

## 5 Concluding Remarks

**Limitations.**    While our study offers valuable insights into the performance of our proposed method across various datasets, we acknowledge certain limitations that open avenues for future exploration and optimization. First, our analysis concentrated on a selection of data heterogeneity scenarios, encompassing natural and partitioned heterogeneity. We acknowledge that our findings may not fully encapsulate the dynamics present in more extreme non-iid scenarios. In fact, for unusually distance local datasets, we do not believe that our proposal would offer performance improvements. In Section **??**, we find that placing additional emphasis on initial model success results in greater personalized performance. Our intuition is based on the

hypothesis that a shared model that performs well for all users and can fine-tune well is often a better initialization that a model that fine-tunes well but may have no initial model success. When users are extremely difference, it will not be possible to find a model that has any ability to perform well for all users. In this case, we expect our method to suggest using $\beta = 1$ (no decay) and recover the FedAvg solution.

Next, we focus mainly on exponential decay to show that scaling local gradient updates allows for balancing the objectives of initial model success and the ability to personalize rapidly. However, many alternative decay schemes for learning rates exist, but we have only briefly explored linear decay in Appendix B.2. Future research may explore these alternative decay schemes to enrich our understanding and potentially enhance our method's performance.

Finally, we evaluated our proposed method in isolation without extensively exploring its compatibility with other state-of-the-art techniques. While this approach enabled us to assess our method's intrinsic merits, practical deployment often requires combining multiple strategies for optimal results. Much of our theoretical work relies on stochastic gradient descent as the optimizer, and we intend to expand our theory to alternative optimizers in future work.

**Conclusion.** In summary, this study introduces an unexplored approach to federated learning by incorporating within-round learning rate decay to balance the objectives of initial model success and rapid personalization. This additional flexibility allows FedDecay to adapt, less decay for more considerable user differences, to the problem-specific data heterogeneity for better performance without changing the rate of convergence. In Section **??**, we consistently demonstrate that some within-round learning rate decay outperforms both no emphasis on rapid personalization (FedSGD) and no decay (FedAvg). In addition to consistently better performance than the popular FedAvg procedure, we often return similar metrics to personalized methods that require additional memory and computation, especially when generalizing to new users. Notably, within-round learning rate decay does not require additional memory and computationally requires only a simple grid search to find improved solutions. In summary, we propose a cheap, interpretable modification to FedAvg that consistently returns a 1-4 percent increase in average test set accuracies.

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

# A  Supplementary Proofs

## A.1  Balancing Initial Model Success and Rapid Personalization

We extend the work of Nichol et al. (2018) to understand how the updates of FedDecay impact initial model success, rapid personalization, and generalization. Consider the gradient from a sequence of loss functions $\{F_i^{(j)}\}_{j=0}^{K-1}$ where $K \in \{z \in \mathbb{Z} \mid z \geq 2\}$. For example, the above sequence could be a local objective function evaluated on different mini-batches. Consider the following update for any sequence of scaling coefficients $\{\beta_j \mid \beta_0 \neq 0\}_{j=0}^{K-1}$.

$$\theta_i^{(k)} = \theta_i^{(k-1)} - \eta\beta_{k-1} \times \nabla F_i^{(k-1)}(\theta_i^{(k-1)}) \text{ for } k = 1, \ldots, K$$

Let $g_i^{(j)} = \nabla F_i^{(j)}(\theta_i^{(j)})$ and define $\tilde{g}_i^{(j)} = \nabla F_i^{(j)}(\theta_g)$ and $\tilde{H}_i^{(j)} = \nabla^2 F_i^{(j)}(\theta_g)$ as the gradient and Hessian of the $j$-th loss function evaluated at the initial point.

$$\begin{aligned}
g_i^{(j)} = \nabla F_i^{(j)}(\theta_i^{(j)}) &\approx \nabla F_i^{(j)}(\theta_g) + \nabla^2 F_i^{(j)}(\theta_g)\left(\theta_i^{(j)} - \theta_g\right) \\
&= \tilde{g}_i^{(j)} + \tilde{H}_i^{(j)}\left(\theta_i^{(j)} - \theta_g\right) \\
&= \tilde{g}_i^{(j)} - \eta\tilde{H}_i^{(j)} \sum_{h=0}^{j-1} \beta_h \nabla F_i^{(h)}(\theta_i^{(h)}) \\
&\approx \tilde{g}_i^{(j)} - \eta\tilde{H}_i^{(j)} \sum_{h=0}^{j-1} \beta_h \tilde{g}_i^{(h)}
\end{aligned}$$

Apply the previous expansion to the update of FedDecay.

$$\begin{aligned}
g_{FedDecay} = \frac{\theta_i^{(K)} - \theta_g}{-\eta} &= \sum_{j=0}^{K-1}\left(\frac{\theta_i^{(j+1)} - \theta_i^{(j)}}{-\eta}\right) \\
&= \sum_{j=0}^{K-1} \beta_j \nabla F_i^{(j)}(\theta_i^{(j)}) \\
&\approx \sum_{j=0}^{K-1} \beta_j\left(\tilde{g}_i^{(j)} - \eta\tilde{H}_i^{(j)} \sum_{h=0}^{j-1} \beta_h \tilde{g}_i^{(h)}\right) \\
&= \sum_{j=0}^{K-1} \beta_j \tilde{g}_i^{(j)} - \eta \sum_{j=0}^{K-1}\left(\beta_j\tilde{H}_i^{(j)} \sum_{h=0}^{(j-1)} \beta_h \tilde{g}_i^{(h)}\right)
\end{aligned}$$

$$\begin{aligned}
\mathbb{E}\left[g_{FedDecay}\right] &\approx \sum_{j=0}^{K-1} \beta_j \tilde{g}_i^{(j)} - \eta \sum_{j=0}^{K-1}\left(\beta_j\tilde{H}_i^{(j)} \sum_{h=0}^{j-1} \beta_h \tilde{g}_i^{(h)}\right) \\
&= \mathbb{E}\left[\tilde{g}_i^{(j)}\right]\left(\sum_{j=0}^{K-1} \beta_j\right) - \mathbb{E}\left[\tilde{H}_i^{(j)}\tilde{g}_i^{(h)}\right]\left(\eta\sum_{j=0}^{K-1}\left(\beta_j\sum_{h=0}^{j-1}\beta_h\right)\right)
\end{aligned}$$

Hence the following ratio of $\mathbb{E}\left[\tilde{H}_i^{(j)}\tilde{g}_i^{(h)}\right]$ to $\mathbb{E}\left[\tilde{g}_i^{(j)}\right]$ for FedDecay. Let $\mathcal{B}(k) = \sum_{j=0}^{k-1} \beta_{j-1}$.

$$R_{FedDecay} = \frac{\eta\sum_{j=0}^{K-1} \beta_j\mathcal{B}(j)}{\mathcal{B}(K)}$$

Note that these equations do not need to be within-round decay and that some expressions hold more generally. However, our interest is in addressing our concern that existing meta-learning techniques, which facilitate rapid personalization, do not require a model to have good initial success.

Exponential decay $\beta_j = \beta^j$ allows for the simplification of the above ratio after using the finite sum formula for the geometric series, $\mathcal{B}(k) = \frac{1-\beta^k}{1-\beta}$.

$$
\begin{aligned}
R_{FedDecay} &= \eta \left( \sum_{j=0}^{K-1} \beta^j \left( \frac{1-\beta^j}{1-\beta} \right) \right) \times \left( \frac{1-\beta}{1-\beta^K} \right) \\
&= \eta\beta \left( \frac{(1-\beta^K)(1-\beta^{K-1})}{(1-\beta)^2(1+\beta)} \right) \times \left( \frac{1-\beta}{1-\beta^K} \right) \\
&= \eta\beta \left( \frac{1-\beta^{K-1}}{(1-\beta)(1+\beta)} \right) \\
&= \eta\beta \left( \frac{1-\beta^{K-1}}{1-\beta^2} \right)
\end{aligned}
$$

## A.2 Convergence of FedDecay on Heterogenous Data

**Full User Participation.** We expand the theoretical work of Li et al. (2020c). Furthermore, we adopt the following additional notations and use their Lemma 1 and Lemma 2. Let iteration $t = nK+1$ denote the $k$-th local update step of communication round $n+1$ of federated training. The $i$-th users model at iteration $t$ is provided by $w_t^i$. Hence, all previous notations can be converted similarly to $w_t^i = \theta_i^{\lfloor t/K \rfloor, \, t \bmod K}$.

Recall that $S \subseteq C$ denotes the set of users participating in the update of the global model for a given round $n$ of federated training. Here, $S = C$ and let $p_i$ denote the aggregation weight of the $i$-th user as introduced in Section 3.2. We are most interested in assigning equal probability or aggregation weight to all users since our objective is an improved initialization (for all users).

$$
\begin{aligned}
v_{t+1}^i &= w_t^i - \eta_t \nabla \hat{F}_i(w_t^i, \xi_t^i) \\
w_{t+1}^i &= \begin{cases} v_{t+1}^i & \text{if } (t+1) \bmod K \neq 0 \\ \sum_{i=1}^{M} p_i v_{t+1}^i & \text{else} \end{cases}
\end{aligned}
$$

Also, we need to define the following aggregated sequences, which are always equivalent under full user participation. Furthermore, shorthand notation for various gradients is used for simplicity. Note $\bar{w}_t$ is only accessible when $t \bmod K = 0$, $\bar{g}_t = \mathbb{E}g_t$, and $\bar{v}_{t+1} = \bar{w}_t - \eta_t g_t$.

$$
\bar{v}_t = \sum_{i=1}^{M} p_i v_t^i \qquad\qquad g_t = \sum_{i=1}^{M} p_i \nabla F_i(w_t^i, \xi_t^i)
$$

$$
\bar{w}_t = \sum_{i=1}^{M} p_i w_t^i \qquad\qquad \bar{g}_t = \sum_{i=1}^{M} p_i \nabla F_i(w_t^i)
$$

We take the following two lemmas from Li et al. (2020c) and adopt the third to our learning rate scheduling.

**Lemma 1.** *Assume Assumption 1 and 2. If $\eta_t \leq \frac{1}{4L}$, we have*

$$
\mathbb{E} \left\| \bar{v}_{(t+1)} - w^* \right\|^2 \leq (1 - \eta_t) \mathbb{E} \left\| \bar{w}_t - w^* \right\|^2 + \eta_t^2 \mathbb{E} \left\| g_t - \bar{g}_t \right\|^2
$$

$$
+ 2\mathbb{E} \left[ \sum_{i=1}^{M} p_i \left\| \bar{w}_t - w_i^t \right\|^2 \right] + 6L\eta_t^2 \Gamma
$$

**Lemma 2.** *Under Assumption 3, it follows that*

$$
\mathbb{E} \left\| g_t - \bar{g}_t \right\|^2 \leq \sum_{i=1}^{M} p_i^2 \sigma_i^2
$$

**Lemma 3.** *Under Assumption 4, a locally decaying, cyclic learning rate of the form $\eta_t = \alpha_{\lfloor t/K \rfloor} \beta_{t \bmod K}$ such that $\beta_{t+1} \leq \beta_t$ for $t = 0, \ldots, K-1$ satisfies*

$$E \left[ \sum_{i=1}^{M} p_i \left\| \bar{w}_t - w_t^i \right\|^2 \right] \leq \eta_t^2 \left( G(K-1)\beta_{K-1}^{-1} \right)^2$$

*Proof.* Note that for all $t \geq 0$ there exists $t_0 \leq t$ such that $t - t_0 \leq K-1$ and $w_{t_0}^i = \bar{w}_{t_0}$ for all $i \in [1, \ldots, M]$

$$\mathbb{E} \left[ \sum_{i=1}^{M} p_i \left\| \bar{w}_t - w_t^i \right\|^2 \right]$$

$$= \mathbb{E} \left[ \sum_{i=1}^{M} p_i \left\| \left( w_t^i - \bar{w}_{t_0} \right) - \left( \bar{w}_t - \bar{w}_{t_0} \right) \right\|^2 \right]$$

$$\leq \mathbb{E} \left[ \sum_{i=1}^{M} p_i \left\| \left( w_t^i - \bar{w}_{t_0} \right) \right\|^2 \right]$$

as $\mathbb{E} \left\| X - \mathbb{E} X \right\| \leq \mathbb{E} \left\| X \right\|^2$ where $X = w_t^i - \bar{w}_{t_0}$

$$\leq \sum_{i=1}^{M} p_i \mathbb{E} \left[ (K-1) \sum_{i=t_0}^{t-1} \eta_t^2 \left\| \nabla \hat{F}_i \left( w_t^i, \xi_t^i \right) \right\|^2 \right]$$

as $\left\| \sum_{i=t_0}^{t-1} \eta_t \nabla \hat{F}_i \left( w_t^i, \xi_t^i \right) \right\|^2 \leq (t - t_0) \sum_{i=t_0}^{t-1} \eta_t^2 \left\| \nabla \hat{F}_i \left( w_t^i, \xi_t^i \right) \right\|^2$

$$\leq (K-1) \sum_{i=1}^{M} p_i \sum_{i=t_0}^{t-1} \eta_{t_0}^2 G^2$$

as $\mathbb{E} \left\| \nabla \hat{F}_i \left( w_t^i, \xi_t^i \right) \right\|^2 \leq G^2$ and $\eta_t \leq \eta_{t_0}$ for $t_0 \leq t \leq t_0 + K$

$$\leq (K-1) \sum_{i=1}^{M} p_i \sum_{i=t_0}^{t-1} \left( \frac{G\beta_0}{\beta_{K-1}} \right)^2$$

as $\eta_{t_0} \leq \eta_t \left( \frac{\beta_0}{\beta_{K-1}} \right)$ for $t_0 \leq t \leq t_0 + K$

$$\leq \eta_t^2 \left( G(K-1)\beta_0 \beta_{K-1}^{-1} \right)^2 \text{ as } \sum_{i=1}^{M} p_i = 1$$

$\square$

The following proves convergence under full user participation $S. = C$.

*Proof.* Recall that for all $t$, under full user participation $\bar{w}_t = \bar{v}_t$. Let $\Delta_t = \mathbb{E} \left\| \bar{w}_t - w^* \right\|^2$. From Lemma 1, 2, and 3, it follows that

$$\Delta_{t+1} \leq \left( 1 - \eta_t \mu \right) \Delta_t + \eta_t^2 B$$

$$\text{where } B = \sum_{i=1}^{M} p_i^2 \sigma_i^2 + 6L\Gamma + 2 \left( G(K-1)\beta_0 \beta_{K-1}^{-1} \right)^2$$

We can assume without loss of generality that $\beta_{t \bmod K} \leq 1$ and $\beta_0 = 1$. Otherwise

$$\eta_t = \alpha_{\lfloor t/K \rfloor} \beta_{t \bmod K} = \tilde{\alpha}_{\lfloor t/K \rfloor} \tilde{\beta}_{t \bmod K}$$

where $\tilde{\alpha}_t = \beta_0 \left(\frac{c}{t+d}\right)$ and $\tilde{\beta}_t = \frac{\beta_t}{\beta_0} \leq 1$. Additionally, we can assume that all $\beta$'s are positive-valued as we could reduce $K$ until this is satisfied.

For $\eta_t = \alpha_{\lfloor t/K \rfloor} \beta_{t \bmod K}$ where $\alpha_t = \frac{c}{t+d}$ for some $c > \frac{2}{\mu\beta_{K-1}}$ and $d > 2 - \frac{1}{K}$ such that $\eta_1 \leq \frac{1}{4L}$.

$$\frac{c}{(t/K)+d} \leq \alpha_{\lfloor t/K \rfloor} \leq \frac{c}{(t/K)+d-1}$$

$$\Rightarrow \frac{c\beta_{K-1}}{(t/K)+d} \leq \eta_t \leq \frac{c}{(t/K)+d-1}$$

$$\text{as } \beta_{K-1} \leq \beta_{t \bmod K} \leq 1$$

By induction, we prove $\Delta_t \leq \dfrac{v}{(t/K)+d-2}$ where

$$v = \max\left\{ \frac{c^2 B}{\beta_{K-1}c\mu - 2}, [(1/K)+d-2]\,\Delta_1 \right\}$$

Note that $t = 1$ holds trivially by the definition of $v$. Assuming the conclusion holds for some $t$, then

$$\Delta_{t+1} \leq (1 - \eta_t\mu)\,\Delta_t + \eta_t^2 B$$

$$\leq \left(1 - \frac{\beta_{K-1}c\mu}{(t/K)+d}\right)\left(\frac{v}{(t/K)+d-2}\right) + \left(\frac{c}{(t/K)+d-1}\right)^2 B$$

by the induction hypothesis, $1 - \eta_t\mu \leq 1 - \dfrac{\beta_{K-1}c\mu}{(t/K)+d}$,

and $\eta_t \leq \left(\dfrac{c}{(t/K)+d-1}\right)$

$$\leq \left(1 - \frac{\beta_{K-1}c\mu}{(t/K)+d}\right)\left(\frac{v}{(t/K)+d-2}\right) + \frac{c^2 B}{[(t/K)+d][(t/K)+d-2]}$$

$$\text{as } (t/K)+d-2 > (1/K)+2-(1/K)-2 = 0$$

$$\Rightarrow [(t/K)+d][(t/K)+d-2] \leq [(t/K)+d-1]$$

Letting $a = \dfrac{1}{[(t/K)+d][(t/K)+d-2]}$, we continue with the previous quantity.

$$= a\left([(t/K)+d-2]\,v + c^2 B - [c\mu\beta_{K-1}-2]\,v\right)$$

$$\leq a\left([(t/K)+d-2]\,v + c^2 B - [c\mu\beta_{K-1}-2]\left[\frac{c^2 B}{\beta_{K-1}c\mu}\right]\right)$$

$$\text{as } \beta_{K-1}c\mu - 2 > 0 \text{ and } v \leq \frac{c^2 B}{\beta_{K-1}c\mu}$$

$$= \frac{[(t/K)+d-2]\,v}{[(t/K)+d][(t/K)+d-2]}$$

$$= \frac{v}{(t/K)+d}$$

$$\leq \frac{v}{(t/K)+d}\left(\frac{(t/K)+d}{(t/K)+d-\left(2+\frac{1}{K}\right)}\right)$$

$$= \frac{v}{(t/K)+d-\left(2+\frac{1}{K}\right)}$$

$$= \frac{v}{\left(\frac{t+1}{K}\right)+d-2}$$

Then by $L$-smoothness of $F(\cdot)$

$$\mathbb{E}\left[F(\bar{w}_t) - F^*\right] \leq \frac{L}{2}\Delta_t \leq \frac{L}{2}\left(\frac{v}{(t/K) + d - 2}\right)$$

Specifically, if we choose the constants to be set as:
$c = \dfrac{3}{\mu\beta_{K-1} - 2}$ and $d = \max\left\{\dfrac{12L}{\mu\beta_{K-1}}, 4 - \dfrac{2}{K}\right\}$ then,

$$\alpha_1 = \frac{c}{d+1} < \frac{c}{d} = \frac{3}{\mu\beta_{K-1}d} \leq \frac{3}{12L + \mu\beta_{K-1}} \leq \frac{1}{4L}$$

$$\text{as } d \geq \frac{12L}{\mu\beta_{K-1}}$$

$$v = \max\left\{\frac{c^2 B}{\beta_{K-1}c\mu - 2}, \left[(1/K) + d - 2\right]\Delta_1\right\}$$

$$\leq \frac{c^2 B}{\beta_{K-1}c\mu - 2} + \left[(1/K) + d - 2\right]\Delta_1$$

$$= c^2 B + \left[(1/K) + d - 2\right]\Delta_1$$

$$\text{as } c = \frac{3}{\mu\beta_{K-1}}$$

$$= \left(\frac{3}{\mu\beta_{K-1}}\right)^2 B + \left[(1/K) + d - 2\right]\Delta_1$$

$$= \frac{2}{\mu}\left(\frac{9B}{2\mu\beta_{K-1}^2} + \frac{(1/K) + d - 2}{2}\Delta_1\right)$$

$$\mathbb{E}\left[F(\bar{w}_t) - F^*\right] \leq \frac{L}{2}\left(\frac{v}{(t/K) + d - 2}\right)$$

$$\leq \frac{\kappa}{(t/K) + d - 2}\left(\frac{9B}{2\mu\beta_{K-1}^2} + \frac{(1/K) + d - 2}{2}\Delta_1\right)$$

$\square$

**Partial User Participation.** Here, we focus on the case where the random set of users at iteration $t$ $(S_t)$ of size $|S|$ is selected to update the global model. Consider when the central server forms $S.$ by sampling uniformly without replacement. We modify the definition of $w_t$ to incorporate the new averaging scheme.

$$w_{t+1}^i = \begin{cases} v_{t+1}^i & \text{if } (t+1) \bmod K \neq 0 \\ \sum_{i \in S_{t+1}} p_i \frac{M}{|S|} v_{t+1}^i & \text{else} \end{cases}$$

We rely on the following previous work by Li et al. (2020c).

**Lemma 4.** *If $t + 1 \bmod K = 0$ and $S.$ is sampled uniformly without replacement, then*

$$\mathbb{E}_{S_t}\left(\bar{v}_{t+1}\right) = \bar{v}_{t+1}$$

**Lemma 5.** *If $t + 1 \bmod K = 0$, $S.$ is sampled uniformly without replacement, $p_i = \frac{1}{n}$ for all $i \in [1, \ldots, M]$, and $\eta_t = \alpha_{\lfloor t/K \rfloor}\beta_{t \bmod K}$ where both $\alpha_j$ and $\beta_j$ are non-increasing, then*

$$\mathbb{E}_{S_t}\left\|\bar{v}_{t+1} - \bar{w}_{t+1}\right\|^2 \leq \eta_t^2\left(\frac{M - |S_t|}{K(M - 1)}\right)\left(GK\beta_0\beta_{K-1}\right)^2$$

*Proof.* Replace $\eta_{t_0} \leq 2\eta_t$ with $\eta_{t_0} \leq \eta_t\left(\frac{\beta_0}{\beta_{K-1}}\right)$ in the original proof. $\square$

The following proves convergence under partial user participation in updating the global model. This proof completes Theorem 1.

*Proof.*

$$
\begin{aligned}
\mathbb{E}_{S_{t+1}} \left\| \bar{w}_{t+1} - w^* \right\|^2 & \\
&= \mathbb{E}_{S_{t+1}} \left\| \bar{w}_{t+1} - \bar{v}_{t+1} + \bar{v}_{t+1} - w^* \right\|^2 \\
&= \mathbb{E}_{S_{t+1}} \left\| \bar{w}_{t+1} - \bar{v}_{t+1} \right\|^2 \\
&\quad + \mathbb{E}_{S_{t+1}} \left\| \bar{v}_{t+1} - w^* \right\|^2 + 2\mathbb{E}_{S_{t+1}} \langle \bar{w}_{t+1} - \bar{v}_{t+1}, \bar{v}_{t+1} - w^* \rangle \\
&= \mathbb{E}_{S_{t+1}} \left\| \bar{w}_{t+1} - \bar{v}_{t+1} \right\|^2 + \mathbb{E}_{S_{t+1}} \left\| \bar{v}_{t+1} - w^* \right\|^2
\end{aligned}
$$

The last equality follows from Lemma 4. Next, by Lemma 5, we have

$$
\mathbb{E} \left\| \bar{w}_{t+1} - w^* \right\|^2 \leq
\begin{cases}
(1 - \eta_t \mu) \mathbb{E} \left\| \bar{w}_t - w^* \right\|^2 + \eta_t B \\
\quad \text{if } t + 1 \bmod K \neq 0 \\
(1 - \eta_t \mu) \mathbb{E} \left\| \bar{w}_t - w^* \right\|^2 + \eta_t (B + D) \\
\quad \text{if } t + 1 \bmod K = 0
\end{cases}
$$

Furthermore, the second bound holds for all $t$ since $D > 0$. Next, similar to our proof under full user participation:

- Assume without loss of generality that $\beta_{t \bmod K} \leq 1$ and $\beta_0 = 1$.

- $\eta_t = \alpha_{\lfloor t/K \rfloor} \beta_{t \bmod K}$ where $\alpha_j = \frac{c}{t+d}$ for some $c > \frac{2}{\mu \beta_{K-1}}$ and $d > 2 - \frac{1}{K}$.

Then $\Delta_t \leq \frac{v}{(t/K) + d - 2}$ where $v = \max \left\{ \frac{c^2(B+D)}{\beta_{K-1} c \mu - 2}, [(1/K) + d - 2] \Delta_1 \right\}$.

Hence $\mathbb{E}\left[ F(\bar{w}_t) \right] - F^* \leq \frac{L}{2} \Delta_t \leq \frac{L}{2} \left( \frac{v}{(t/K) + d - 2} \right)$.

With $c = \frac{3}{\mu \beta_{K-1}}$ and $d = \max \left\{ \frac{12L}{\mu \beta_{K-1}}, 4 - \frac{2}{K} \right\}$,

$$
\mathbb{E}\left[ F(\bar{w}_t) - F^* \right] \leq \frac{\kappa}{(t/K) + d - 2} \left( \frac{9(B+D)}{2\mu \beta_{K-1}^2} + \left( \frac{(1/K) + d - 2}{2} \right) \Delta_1 \right)
$$

$\qquad\qquad\qquad\qquad\qquad\qquad\qquad\qquad\qquad\qquad\qquad\qquad\qquad\qquad\qquad\qquad\qquad$ □

## B    Supplementary Results

### B.1    Sensitivity Analysis of FedDecay

FedDecay demonstrates sensitivity to the hyper-parameter $\beta$ choice; see Table 5. Specifically, in the case of the FEMNIST data set, runs with $\beta = 0.8$ were prematurely halted due to hyperband stopping, indicating inferior performance compared to smaller $\beta$ choices. Conversely, the remaining $\beta$ values improved accuracy for new and existing users. Notably, the FEMNIST data set, comprised of handwritten characters contributed by different authors, showcases inherent data heterogeneity. This real-world example underscores the presence of substantial user similarities, owing to the shared language in which all characters are written.

This trend aligns with our assertion in Section 3.2, where we indicated that prioritizing AvgGrad terms (via smaller $\beta$ values) would be beneficial in cases where users share similarities. Consequently, we observe FedSGD outperforming FedAvg in this context. However, FedSGD, which does not emphasize AvgGradInner

| Method | $\beta$ | FEMNIST (image) | | SST2 (text) | | PUBMED (graph) | |
|---|---|---|---|---|---|---|---|
| | | Existing $\bar{Acc}$ | New $\bar{Acc}$ | Existing $\bar{Acc}$ | New $\bar{Acc}$ | Existing $\bar{Acc}$ | New $\bar{Acc}$ |
| FedSGD | 0.0 | 88.51 | 90.55 | 73.60 | 80.30 | 86.70 | 79.14 |
| FedDecay | 0.2 | **89.86** | **91.52** | 67.01 | 59.02 | 85.67 | 79.68 |
| FedDecay | 0.4 | 89.74 | 90.77 | 69.42 | 67.26 | **87.22** | **80.39** |
| FedDecay | 0.6 | 89.76 | 90.93 | **78.15** | **81.01** | 86.65 | 79.86 |
| FedDecay | 0.8 | Hyperband Stopped | | 74.64 | 76.14 | 86.59 | 79.32 |
| FedAvg | 1.0 | 87.91 | 89.78 | 76.54 | 76.80 | 86.29 | 79.50 |

Table 5: Performance of FedDecay under misspecified values for decay coefficient, $\beta$.

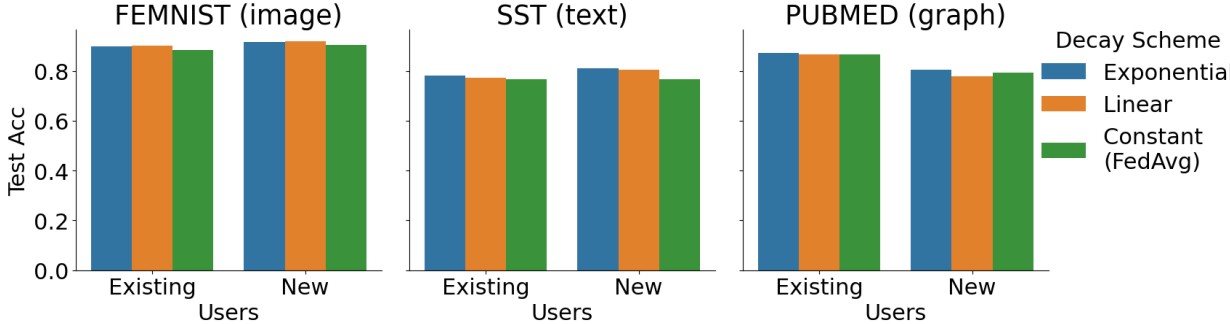

Figure 2: Alternative Without-Round Decay Scheme Performance. Linear decay scheme can also provide improved performance over FedAvg. Within-round decay in general is an option for improving performance without additional computation or memory cost.

(generalization, rapid personalization), is beaten by FedDecay. Here, we observe that having a more flexible balance on initial model success, generalization, and fast personalization results in better performance for various choices of $\beta$.

Unlike FEMNIST, heterogeneous users are created from SST2 by partitioning data into 50 clients using Dirichlet allocation with $\alpha = 0.4$. Many extreme non-iid data sets exist, but $\alpha = 0.4$ results in reasonably non-iid users. Revisiting the claim discussed in our Section 3.2, we anticipate that an increased emphasis (achieved via larger $\beta$ values) on AvgGradInner is required as non-iid user diversity grows. In general, larger values of $\beta$ perform better on SST2. While FedSGD demonstrates superior test-set performance compared to FedAvg, its validation-set accuracy lags behind FedAvg's, preventing it from securing the best run on SST2 for FedAvg in Section 4.1. More importantly, FedDecay balances AvgGrad and AvgGradInner terms, yielding significantly improved metrics over FedAvg. Notably, FedDecay still has the best overall performance among the three methods. Lastly, in the case of the PUBMED data set, our method's performance seems resilient to the choice of $\beta$. However, a straightforward grid search enhances performance for new and existing users over FedAvg and FedSGD.

## B.2 Linear Within-Round Learning Rate Decay

In this section, we explore the potential of alternative decay schemes within the FedDecay framework. Results in Section 3.2 allow for more general sequences than exponential decay. While exponential decay has been a focus, we believe other strategies for scaling gradient emphasis during federated training could enhance performance by their balance between initial model success, generalization, and rapid personalization. We expand our analysis to include the use of linear decay for FedDecay. We define linear decay as $\beta_j = \max\{1 - j(1 - \beta), 0\}$, where $1 - \beta$ is employed for consistency aligning with FedSGD ($\beta = 0$) and FedAvg ($\beta = 1$). Figure 2 illustrates the average test set accuracy for new and existing users across different decay schemes.

It is evident from the results that both exponential and linear decay schemes yield performance improvements over FedAvg. While linear decay performs slightly worse than FedAvg on the new PUBMED user, it demonstrates robust performance on FEMNIST and SST2. These findings underscore the potential for alternative decay schemes to enhance the overall model performance within the federated learning paradigm. We emphasize that decay can be an effective tool for improving performance without additional cost.

## B.3   Robustness of FedDecay to Choice of Seed

We address conducting a series of experiments to ensure the robustness of our findings against the selection of random seeds. Given the constraints of our computational resources, we focus on replicating the less computational FL methods using three distinct fixed seeds. We chose FEMNIST and PUBMED to have one example of full and partial user participation. We present the averaged top-run metrics for new and existing users across the FEMNIST and PUBMED data sets in Table 6 and Table 7. Notably, the FedDecay approach consistently achieves the highest average accuracy among the FL algorithms, catering to new and existing users. Moreover, it demonstrates exceptional performance regarding the best ten-percentile accuracy for existing users. This reinforces the reliability and generalizability of our proposed FedDecay method.

| Method | Existing Users | | | New Users | | |
| | $\bar{Acc}$ | $\breve{Acc}$ | $\sigma_{Acc}$ | $\bar{Acc}$ | $\breve{Acc}$ | $\sigma_{Acc}$ |
|---|---|---|---|---|---|---|
| FOMAML | 0.8971 | 0.8223 | 0.0687 | 0.9044 | **0.8371** | 0.0578 |
| FedAvg | 0.8948 | 0.8227 | 0.0701 | 0.9095 | 0.8318 | 0.0563 |
| **FedDecay** | **0.8993** | **0.8285** | 0.0688 | **0.9127** | 0.8294 | 0.0545 |

Table 6:   Generalization metrics for new and existing users on the FEMNIST (image) data set. Values reported are the average metric for each single-model-based method's top run across several seeds. FedDecay produces the maximum average test set accuracy on new and existing users.

| Method | Existing Users | | | New Users | | |
| | $\bar{Acc}$ | $\breve{Acc}$ | $\sigma_{Acc}$ | $\bar{Acc}$ | $\breve{Acc}$ | $\sigma_{Acc}$ |
|---|---|---|---|---|---|---|
| FOMAML | 0.8597 | 0.8234 | 0.0293 | 0.7855 | - | - |
| FedAvg | 0.8666 | 0.8313 | 0.0328 | 0.7914 | - | - |
| **FedDecay** | **0.8704** | **0.8420** | 0.0244 | **0.7950** | - | - |

Table 7:   Generalization metrics for new and existing users on the PUBMED (graph) data set. Values reported are the average metric for each single-model-based method's top run across several seeds. FedDecay produces the maximum average test set accuracy on new and existing users. With five total users, only a single user is held out to evaluate generalization. Hence, there are no values for the bottom ten percentile ($\breve{Acc}$) or standard deviation ($\sigma_{Acc}$) for new users.

## B.4   Method Hyper-parameters After Tuning

We provide the hyperparameter configurations for all methods that produce the top average validation set accuracy. These hyperparameter configurations are then used to compute test set metrics for our main experiments.

| Method | Local Epochs ($K$) | Batch Size | Learning Rate ($\alpha$) | Regularization Rate | Meta-learning Steps | Local Decay ($\beta$) |
|---|---|---|---|---|---|---|
| Ditto | 3 | - | 0.10 | 0.50 | - | - |
| FedBN | 3 | - | 0.01 | - | - | - |
| FedEM | 3 | - | 0.10 | - | - | - |
| pFedMe | 3 | - | 0.50 | 0.05 | 3.0 | - |
| FOMAML | 3 | - | 0.50 | - | - | - |
| FedAvg | 1 | - | 0.10 | - | - | - |
| FedDecay - Exponential | 3 | - | 0.05 | - | - | 0.2 |
| FedDecay - Linear | 3 | - | 0.10 | - | - | 0.6 |

Table 8: Main experiment hyper-parameters for all methods after tuning on FEMNIST

| Method | Local Epochs ($K$) | Batch Size | Learning Rate ($\alpha$) | Regularization Rate | Meta-learning Steps | Local Decay ($\beta$) |
|---|---|---|---|---|---|---|
| Ditto | 3 | 16.0 | 0.05 | 0.8 | - | - |
| FedBN | 3 | 64.0 | 0.10 | - | - | - |
| FedEM | 3 | 16.0 | 0.05 | - | - | - |
| pFedMe | 3 | 64.0 | 0.05 | 0.8 | 3.0 | - |
| FOMAML | 3 | 16.0 | 0.05 | - | - | - |
| FedAvg | 3 | 16.0 | 0.05 | - | - | - |
| FedDecay - Exponential | 3 | 16.0 | 0.05 | - | - | 0.6 |
| FedDecay - Linear | 3 | 16.0 | 0.05 | - | - | 0.4 |

Table 9: Main experiment hyper-parameters for all methods after tuning on SST

| Method | Local Epochs ($K$) | Batch Size | Learning Rate ($\alpha$) | Regularization Rate | Meta-learning Steps | Local Decay ($\beta$) |
|---|---|---|---|---|---|---|
| Ditto | 1 | - | 0.005 | 0.1 | - | - |
| FedBN | 3 | - | 0.005 | - | - | - |
| FedEM | 3 | - | 0.500 | - | - | - |
| pFedMe | 3 | - | 0.500 | 0.5 | 1.0 | - |
| FOMAML | 3 | - | 0.500 | - | - | - |
| FedAvg | 1 | - | 0.500 | - | - | - |
| FedDecay - Exponential | 3 | - | 0.500 | - | - | 0.4 |
| FedDecay - Linear | 3 | - | 0.500 | - | - | 0.4 |

Table 10: Main experiment hyper-parameters for all methods after tuning on PUBMED

