# OpenReview forum: "Balancing Model Performance and Rapid Personalization in Federated Learning with Learning Rate Scheduling"
_TMLR — Rejected by TMLR_

### Review · Reviewer_teRv · 2024-11-01

**Summary Of Contributions:**

This paper studies the problem of personalized federated learning and presents a new approach based on learning rate decay for balancing the initial model performance (before personalization) and the personalization performance. The learning rate decay approach multiplies the vanilla learning rate $\eta$ with a $\beta^j$ decay term. Here, $j$ indexes the local training steps.

**Audience:**

No

**Broader Impact Concerns:**

N/A.

**Claims And Evidence:**

No

**Requested Changes:**

1. Adding precise definitions to each goal.
2. Adding explanations and discussions for the mathematical equations.
3. Adding motivations for the balancing goal and the algorithm design.
4. Comparing the new approach with prior meta-learning federated learning approaches.

**Strengths And Weaknesses:**

**Strength**
1. The goal of balancing the initial model performance and the personalization performance is clear.
2. The experiments have a good coverage across a few domains such as vision, text, and graph.

**Weakness**
1. The goal lacks motivations. Personalized federated learning aims to improve the personalization performance. Obtaining a good initial model may improve the personalization performance. However, balancing the initial model performance and the personalization performance is not necessarily our goal.
2. Several key concepts lack definition. In Section 3, Methodology, the initial model performance and personalization performance are not defined. The authors do define a local objective function $F$ but I do not know when this function $F$ applies to the initial model or the personalized model.
3. The learning rate decay strategy needs motivation. The authors directly cite the analysis in FOMAML (Nichol et al., 2018) approach and claim that "each method gradient is approximately the weighted sum of two other gradient terms: the first promotes
initial model success, and the second contributes towards rapid personalization". However, I do not see how the learning rate decay scheme interacts with the two gradient terms.
4. The discussion on prior work can be more rigorous. This work follows the FOMAML paper on meta-learning. Meta-learning is a different problem and uses a different setup. These differences worthy more discussions. In addition, the related work section mentions a few limitations of prior meta-learning federated learning paper but these limitations are not grounded in the Methodology section.
5. Presentations need improvements. Each equation could use a concise explanation. For example, "Equation 5 gives exponential within-round learning rate decay, which eads to Equation 6, the method gradient for FedDecay" is very hard to parse without explaining Equations 5 and 6.

---

> ### Author Response · Authors · 2024-11-28
> **Response to Reviewer teRv (1/4)**
>
> Weakness:
>
> The goal lacks motivations. Personalized federated learning aims to improve the personalization performance. Obtaining a good initial model may improve the personalization performance. However, balancing the initial model performance and the personalization performance is not necessarily our goal.
>
> Reply:
>
> We apologize for any confusion. Although we use “balancing,” we mean “finding the right amount of decay to improve personalization performance.” Our extensive experimentation demonstrates that prioritizing initial model success relative to FedAvg and FOMAML leads to improved performance. We believe that an initialization that performs well and can rapidly personalize is often better than an initialization that can rapidly personalize but may not produce meaningful results without fine-tuning. We would be happy to refine our phrasing if you believe it will have a meaningful impact on our readability.
>
> Although not the focus of our work, focusing on personalization alone is inadequate for applications where new users may have insufficient data for fine-tuning. Fine-tuning is an excellent strategy for personalization in federated learning without extra computation or memory requirements. Our proposal should extend the benefits of personalization to applications with new users with little to no data. Hence, our proposal attempts to find an initialization that performs well without needing to be fine-tuned but can also be quickly personalized when data is available. As an additional benefit, when data is available, finding an initialization with good performance increases performance after fine-tuning vs. alternatives like FedAvg and FOMAML.

---

> ### Author Response · Authors · 2024-11-28
> **Response to Reviewer teRv (2/4)**
>
> Weakness and requested change:
>
> - Several key concepts lack definition. In Section 3, Methodology, the initial model performance and personalization performance are not defined. The authors do define a local objective function but I do not know when this function applies to the initial model or the personalized model.
> - Adding precise definitions to each goal.
>
> Reply:
>
> Thank you for the opportunity to clarify. The local objective function can be applied to either the initial model $\theta_g$ or $\theta_i$. We now provide mathematical definitions for “initial model success” and “rapid personalization” at the beginning of our methodology section. See Equations 1 and 2 in the revised paper. Initial model success is the average user local objective function $F_i$ applied to the shared model $\theta_g$ and is commonly minimized in Federated Learning. Similarly, rapid personalization is the average user local objective function applied to the personalized models $\theta_i$ and is frequently minimized in Personalized Federated Learning. Our work shows that placing additional emphasis on initial model success, relative to FedAvg, improves its personalized performance. Please let us know whether the revised work helps make this more clear. We have updated the paper accordingly and colored the updated parts blue for easy reference.

---

> ### Author Response · Authors · 2024-11-28
> **Response to Reviewer teRv (3/4)**
>
> Weaknesses and requested changes:
>
> - The learning rate decay strategy needs motivation. The authors directly cite the analysis in FOMAML (Nichol et al., 2018) approach and claim that "each method gradient is approximately the weighted sum of two other gradient terms: the first promotes initial model success, and the second contributes towards rapid personalization". However, I do not see how the learning rate decay scheme interacts with the two gradient terms.
> - Presentations need improvements. Each equation could use a concise explanation. For example, "Equation 5 gives exponential within-round learning rate decay, which eads to Equation 6, the method gradient for FedDecay" is very hard to parse without explaining Equations 5 and 6.
> - Adding motivations for the balancing goal and the algorithm design.
> - Comparing the new approach with prior meta-learning federated learning approaches.
> - Adding explanations and discussions for the mathematical equations.
>
> Reply:
>
> Thank you for raising these concerns. Please see that we have updated the paper, and the significant changes are colored in blue for easy reference. Many of the changes to our methodology section are intended to improve this specific understanding. Within-round learning rate decay leads to coefficients for each gradient term that depend on $\beta$ the amount of decay in the expected method gradient of FedDecay in Equation 10. As discussed in Table 1, larger values of $\beta$ emphasize rapid personalization (AvgGradInner). Alternative meta-learning solutions Reptile (equivalently FedAvg) and FOMAML can only control the relative emphasis between initial model success and rapid personalization through the number of local update steps $K$. We believe these applications need to sufficiently account for the similarities between ordinary users in FL applications.

---

> ### Author Response · Authors · 2024-11-28
> **Response to Reviewer teRv (4/4)**
>
> Weakness:
>
> The discussion on prior work can be more rigorous. This work follows the FOMAML paper on meta-learning. Meta-learning is a different problem and uses a different setup. These differences worthy more discussions. In addition, the related work section mentions a few limitations of prior meta-learning federated learning paper but these limitations are not grounded in the Methodology section.
>
> Reply:
>
> We hear your concerns and have spent time making some serious revisions to the quality of our presentation. Please see the revised paper with significant changes in blue for easy reference. In addition to the many new references, we would be open to suggestions for other works you think we may need to include.
>
> While we have made significant changes, many works, whether in federated learning, personalized federated learning, or meta-learning, are outside the intended scope of our paper. The lack of recent papers is not due to our engagement but rather the intended direction of our paper. We propose using learning rate scheduling to emphasize initial model success to achieve improved personalized performance.
>
> First, only PFL methods that specifically personalize a single, shared model to all users are directly relevant to the strategy that we propose here. Fair PFL comparisons mainly consist of either FL methods with fine-tuning or meta-learning solutions. Furthermore, while more recent PFL solutions, like KNN-Per [8], may provide a better upper bound on performance, other recent PFL papers [4] only slightly outperform PFL work, like Ditto, included in the benchmark for our experiments.
>
> Second, many meta-learning works, like Per-FedAvg [2], require second-order gradient computations, which would accumulate additional computation not required by our simple proposal. Additionally, Per-FedAvg was a baseline method for the pFedMe paper, which is included in our experiments. Next, some recent meta-learning solutions, like FedABML [1], only show a typical 1 percent improvement over FedAvg with fine-tuning.
>
> Finally, many works exist [1, 3, 6, 7] that conclude that the FedAvg procedure, with fine-tuning, generally outperforms many PFL solutions. Our paper exists to establish that with a simple, interpretable change, we can consistently improve how FedAvg with fine-tuning, one of the essential baselines to PFL, performs on a variety of domains and data heterogeneity supported by our extensive experimentations.
>
> Some, but not all, of the newly included references:
>
> [1] Liu et al., 2023, Personalized Federated Learning via Amortized Bayesian Meta-Learning
>
> [2] Fallah et al., 2020, Personalized Federated Learning: A Meta-Learning Approach
>
> [3] Tan et al., 2022, Towards Personalized Federated Learning
>
> [4] Tavakoli et al. 2024, A Comprehensive View of Personalized Federated Learning on Heterogeneous Clinical Datasets
>
> [5] Marfoq et al., 2022, Personalized Federated Learning through Local Memorization
>
> [6] Matsuda et al., 2024, Benchmark for Personalized Federated Learning
>
> [7] Cheng et al., 2021, Fine-tuning is Fine in Federated Learning
>
> [8] Marfoq et al., 2022, Personalized Federated Learning through Local Memorization

---

### Review · Reviewer_PAMt · 2024-11-04

**Summary Of Contributions:**

#### This work proposes FedDecay, a Personalized Federated learning technique that introduces within-round learning rate decay, equivalent to scaling the magnitude of gradients computed during local training. The learning rate decay allows flexibility between initial model success and rapid personalization. The proposed method reduces the extra computation and memory costs that other PFL methods incur.

**Audience:**

Yes

**Broader Impact Concerns:**

The work does not entail any ethical implications.

**Claims And Evidence:**

Yes

**Requested Changes:**

See the weakness section.

**Strengths And Weaknesses:**

### Strengths
1. An intuitive and cost-effective proposed method. Most PFL methods require extensive computing/memory, additional local models, and much hyper-parameter tuning.
2. Convergence analysis is provided for the proposed method.
3. Experiments cover three diverse domains: vision, text, and graph.

### Weakness
1. I am concerned about how the beta hyperparameter is selected. What's the cost of the search? Is beta the same for all the users, and how much does it impact performance? For example, in Table 5, for the SST2 dataset, the performance drop is about 20% between FedSGD and FedDecay for the 0.2 value.
2. I am also curious to see how FedDecay performs for extreme cases of non-iid.
3. Overall, the proposed method seems very simple and intuitive; I wonder what key challenges the authors faced to enable learning rate decay for PFL.
4. It is also important to add recent related PFL works, for example, KNN-Per (https://arxiv.org/pdf/2111.09360). Most related works authors compared their work to are from the year 2021.

---

> ### Author Response · Authors · 2024-11-28
> **Response to Reviewer PAMt (1/4)**
>
> Weakness:
>
> I am concerned about how the beta hyperparameter is selected. What's the cost of the search? Is beta the same for all the users, and how much does it impact performance? For example, in Table 5, for the SST2 dataset, the performance drop is about 20% between FedSGD and FedDecay for the 0.2 value.
>
> Reply:
>
> The hyperparameter $\beta$ is the same for all users, and we only perform a simple grid search over a few values for $\beta$. Hence, the search cost order (big-O) is the same as that of FedAvg. You correctly point out that the performance is sensitive to the choice of $\beta$, which we acknowledge as a limitation. However, this simple tuning strategy typically shows a 1-4% improvement in average performance over FedAvg, indicating that the simple search is worth finding a solution that better generalizes to new data and new users. Furthermore, with hyperband stopping and early termination, we can further reduce the search cost by terminating runs early that are not showing promising performance. The other methods we compare against, in addition to extra computation and memory usage per run, often have several hyper-parameters they need to tune. For example, pFedMe required tuning its regularization rate and number of meta-learning steps. Our proposal is a simple, intuitive extension of FedAvg that consistently outperforms it across various domains.

---

> ### Author Response · Authors · 2024-11-28
> **Response to Reviewer PAMt (2/4)**
>
> Weakness:
>
> I am also curious to see how FedDecay performs for extreme cases of non-iid.
>
> Reply:
>
> In our experiments, the SST clients are created by Dirichlet partitioning with $\alpha = 0.4$, which is considerable data heterogeneity. We followed the experiment settings in the benchmark paper [1] to have experiments suitable for personalized federated learning. We found on SST that we outperformed the more resource-draining FedEM and pFedME, especially with new users.
>
> However, if local datasets are extremely non-iid, we don’t expect our method to offer an advantage over existing personalized works. Learning a single shared model through Federated Learning works amazing when users are similar, and Personalized Federated Learning works best when users are very different. We believe that most applications have some similarity between users. Hence, our method balances the objectives of FL and PFL and can improve performance on such applications. When local datasets are extremely non-iid, we would expect that hyperparameter tuning would suggest $\beta = 1$, which would make our method equivalent to FedAvg or Reptile. In summary, we expect our method to perform no worse when users are highly non-iid but better when some similarities exist between local datasets. We have added this discussion to our Limitations section for improved transparency. Please see the revised paper with significant changes in blue for easier referencing.
>
> [1] 2022, Chen et al., pFL-Bench: A Comprehensive Benchmark for Personalized Federated Learning

---

> ### Author Response · Authors · 2024-11-28
> **Response to Reviewer PAMt (3/4)**
>
> Weakness:
>
> Overall, the proposed method seems very simple and intuitive; I wonder what key challenges the authors faced to enable learning rate decay for PFL.
>
> Reply:
>
> Our work has a few limitations. Since additional models are not learned during training, there may be better choices than our method when data is highly heterogeneous. However, we may not know whether the datasets are highly heterogeneous or homogenous due to data privacy. Hence, our method is often a good option since it can adapt to various data distributions for improved performance after fine-tuning. Furthermore, since we only have one hyper-parameter ($\beta$), tuning the method for optimal performance is not difficult.

---

> ### Author Response · Authors · 2024-11-28
> **Response to Reviewer PAMt (4/4)**
>
> Weakness:
>
> It is also important to add recent related PFL works, for example, KNN-Per (https://arxiv.org/pdf/2111.09360). Most related works authors compared their work to are from the year 2021.
>
> Reply:
>
> We hear your concerns and have spent time making some serious revisions to the quality of our presentation. Please see the revised paper with significant changes in blue for easy reference. In addition to the many new references, we would be open to suggestions for other works you think we may need to include.
>
> While we have made significant changes, many works, whether in federated learning, personalized federated learning, or meta-learning, are outside the intended scope of our paper. The lack of recent papers is not due to our engagement but rather the intended direction of our paper. We propose using learning rate scheduling to emphasize initial model success to achieve improved personalized performance.
>
> First, only PFL methods that specifically personalize a single, shared model to all users are directly relevant to the strategy that we propose here. Fair PFL comparisons mainly consist of either FL methods with fine-tuning or meta-learning solutions. Furthermore, while more recent PFL solutions, like KNN-Per [8], may provide a better upper bound on performance, other recent PFL papers [4] only slightly outperform PFL work, like Ditto, included in the benchmark for our experiments.
>
> Second, many meta-learning works, like Per-FedAvg [2], require second-order gradient computations, which would accumulate additional computation not required by our simple proposal. Additionally, Per-FedAvg was a baseline method for the pFedMe paper, which is included in our experiments. Next, some recent meta-learning solutions, like FedABML [1], only show a typical 1 percent improvement over FedAvg with fine-tuning.
>
> Finally, many works exist [1, 3, 6, 7] that conclude that the FedAvg procedure, with fine-tuning, generally outperforms many PFL solutions. Our paper exists to establish that with a simple, interpretable change, we can consistently improve how FedAvg with fine-tuning, one of the essential baselines to PFL, performs on a variety of domains and data heterogeneity supported by our extensive experimentations.
>
> Some, but not all, of the newly included references:
>
> [1] Liu et al., 2023, Personalized Federated Learning via Amortized Bayesian Meta-Learning
>
> [2] Fallah et al., 2020, Personalized Federated Learning: A Meta-Learning Approach
>
> [3] Tan et al., 2022, Towards Personalized Federated Learning
>
> [4] Tavakoli et al. 2024, A Comprehensive View of Personalized Federated Learning on Heterogeneous Clinical Datasets
>
> [5] Marfoq et al., 2022, Personalized Federated Learning through Local Memorization
>
> [6] Matsuda et al., 2024, Benchmark for Personalized Federated Learning
>
> [7] Cheng et al., 2021, Fine-tuning is Fine in Federated Learning
>
> [8] Marfoq et al., 2022, Personalized Federated Learning through Local Memorization

---

### Review · Reviewer_i8V6 · 2024-11-16

**Summary Of Contributions:**

This paper explores the use of learning rate decay within each training round in Federated Learning (FL) to balance model performance across diverse local datasets and improve performance after fine-tuning.  The authors do not propose a new Personalized Federated Learning (PFL) method but provide theoretical insights and empirical evidence demonstrating that learning rate scheduling alone can improve generalization to new data.

**Audience:**

Yes

**Broader Impact Concerns:**

Not applied.

**Claims And Evidence:**

Yes

**Requested Changes:**

please see the weaknesses.

**Strengths And Weaknesses:**

Strengths:

1. The paper addresses a crucial problem in FL, which is achieving good model performance across diverse local datasets.

2. The authors provide extensive experimental results that span multiple domains (vision, text, and graph data), lending credibility to the practical utility of their approach.

Weaknesses:

1. There are already numerous works considering learning rate decay in FL, limiting the novelty of this contribution.  The contribution of this paper might be perceived as incremental rather than substantial.

2. The theoretical analysis presented is quite standard and does not offer significant advances over existing FL literature.

3. The references included in the paper are not up-to-date, with the latest citations from 2022. This indicates a potential lack of engagement with the most current research in the field.

4. The reported performance gain of 1 to 4 percentage points in average test set accuracy may be considered limited compared to other advancements in FL and PFL.

5. The related work section lacks clear structure and thoroughness. It does not provide a comprehensive overview of recent developments in the field of FL, which weakens the foundation of the paper.

---

> ### Author Response · Authors · 2024-11-28
> **Response to Reviewer i8V6 (1/4)**
>
> Weakness:
>
> There are already numerous works considering learning rate decay in FL, limiting the novelty of this contribution. The contribution of this paper might be perceived as incremental rather than substantial.
>
> Reply:
>
> Our proposal's simplicity to understand and implement is an advantage for its adoption. We are not aware of other works focusing on within-round learning rate decay. Would you please provide references to such works in federated learning, considering learning rate decay? We would be happy to do further investigation and provide comparisons.

---

> ### Author Response · Authors · 2024-11-28
> **Response to Reviewer i8V6 (2/4)**
>
> Weakness:
>
> The reported performance gain of 1 to 4 percentage points in average test set accuracy may be considered limited compared to other advancements in FL and PFL.
>
> Reply:
>
> Our consistent performance improvement over FedAvg is a meaningful finding. As stated in [1], “standard federated learning methods (e.g., FedAvg) with fine-tuning often outperform personalized federated learning methods.” More recent works related to meta-learning [2] show even more slight differences in performance with the popular FedAvg procedure. Hence, the average improvements in [1, 2, 3, 4] suggest that our 1-4% improvements are not trivial. Our findings indicate that within-round learning rate decay is a meaningful lever for handling data heterogeneity that has yet to be explored.
>
>
> [1] Matsuda et al., 2024, Benchmark for Personalized Federated Learning
>
> [2] Liu et al., 2023, Personalized Federated Learning via Amortized Bayesian Meta-Learning
>
> [3] Chen et al., 2021, On Bridging Generic and Personalized Federated Learning
>
> [4] Cheng et al., 2021, Fine-tuning is Fine in Federated Learning

---

> ### Author Response · Authors · 2024-11-28
> **Response to Reviewer i8V6 (3/4)**
>
> Weakness:
>
> The theoretical analysis presented is quite standard and does not offer significant advances over existing FL literature.
>
> Reply:
>
> Assuming that you are referring to our convergence analysis, we never intended for this to be a critical theoretical contribution to our work. Instead, this result emphasizes that the benefits of within-round learning rate decay justified earlier do not need to come at the expense of algorithmic degradation. This result was intended merely to add confidence to the quality of using our proposal. We have added additional language to emphasize this belief. Please see the revised paper with significant changes in blue for easier referencing.
>
> The main theoretical contribution of our work is justifying that the within-round learning rate can be used as a tool to control the emphasis between the objectives of initial model success and rapid personalization. The main empirical finding, against intuition, is that we achieve solutions with improved personalized performance by emphasizing initial model success more than FedAvg. An initialization with some initial model success and the ability to personalize tends to personalize better than a model that personalizes well but may not exhibit any initial model success. This finding is worthy of publication because we are the first work to focus exclusively on within-round learning rate modification and connect its effect to various aspects of data heterogeneity.

---

> ### Author Response · Authors · 2024-11-28
> **Response to Reviewer i8V6 (4/4)**
>
> Weaknesses:
>
> - The references included in the paper are not up-to-date, with the latest citations from 2022. This indicates a potential lack of engagement with the most current research in the field.
> - The related work section lacks clear structure and thoroughness. It does not provide a comprehensive overview of recent developments in the field of FL, which weakens the paper's foundation.
>
> Reply:
>
> We hear your concerns and have spent time making some serious revisions to the quality of our presentation. Please see the revised paper with significant changes in blue for easy reference. In addition to the many new references, we would be open to suggestions for other works you think we may need to include.
>
> While we have made significant changes, many works, whether in federated learning, personalized federated learning, or meta-learning, are outside the intended scope of our paper. The lack of recent papers is not due to our engagement but rather the intended direction of our paper. We propose using learning rate scheduling to emphasize initial model success to achieve improved personalized performance.
>
> First, only PFL methods that specifically personalize a single, shared model to all users are directly relevant to the strategy that we propose here. Fair PFL comparisons mainly consist of either FL methods with fine-tuning or meta-learning solutions. Furthermore, while more recent PFL solutions, like KNN-Per [8], may provide a better upper bound on performance, other recent PFL papers [4] only slightly outperform PFL work, like Ditto, included in the benchmark for our experiments.
>
> Second, many meta-learning works, like Per-FedAvg [2], require second-order gradient computations, which would accumulate additional computation not required by our simple proposal. Additionally, Per-FedAvg was a baseline method for the pFedMe paper, which is included in our experiments. Next, some recent meta-learning solutions, like FedABML [1], only show a typical 1 percent improvement over FedAvg with fine-tuning.
>
> Finally, many works exist [1, 3, 6, 7] that conclude that the FedAvg procedure, with fine-tuning, generally outperforms many PFL solutions. Our paper exists to establish that with a simple, interpretable change, we can consistently improve how FedAvg with fine-tuning, one of the essential baselines to PFL, performs on a variety of domains and data heterogeneity supported by our extensive experimentations.
>
> Some, but not all, of the newly included references:
>
> [1] Liu et al., 2023, Personalized Federated Learning via Amortized Bayesian Meta-Learning
>
> [2] Fallah et al., 2020, Personalized Federated Learning: A Meta-Learning Approach
>
> [3] Tan et al., 2022, Towards Personalized Federated Learning
>
> [4] Tavakoli et al. 2024, A Comprehensive View of Personalized Federated Learning on Heterogeneous Clinical Datasets
>
> [5] Marfoq et al., 2022, Personalized Federated Learning through Local Memorization
>
> [6] Matsuda et al., 2024, Benchmark for Personalized Federated Learning
>
> [7] Cheng et al., 2021, Fine-tuning is Fine in Federated Learning
>
> [8] Marfoq et al., 2022, Personalized Federated Learning through Local Memorization

---

### Author Response · Authors · 2024-11-28
**Response to Reviewers**

We sincerely thank the reviewers for their initial feedback. We have rewritten considerable portions of the paper with the provided constructive criticism in mind. We provide a revision that is colored in blue wherever significant changes have been made to the original submission, and we look forward to further discussion and feedback. Thank you!

---

### Decision · Action_Editor_KGZ1 · 2025-02-04

**Recommendation:** Reject

**Comment:**

The reviewers still have concerns regarding relations and comparisons to meta-learning methods as well as presentation. The main concern is that the proposed client-side learning rate decay method has been well explored in existing works or federated learning packages/libraries, e.g., https://arxiv.org/pdf/2007.00878, https://jhc.sjtu.edu.cn/~chen-chen/papers/2dlrd-ijcnn21.pdf, https://github.com/google-research/federated/blob/master/adaptive_lr_decay/README.md. Tensorflow Federated supports client and server learning rates to decay independently.

The connections between fedsgd, fedavg, and some meta-learning methods (including maml and reptile) through the lens of varying local step sizes have been investigated and analyzed in previous works, e.g., https://arxiv.org/pdf/2007.00878, which also proposes and evaluates the decaying inner learning rate schedule. The convergence analysis is standard, as acknowledged in the submission as well.

Hence, I suggest the authors to revise the paper significantly conditioned on the existence of prior works on client learning rate decay, and resubmit.

**Audience:**

Yes, the paper is relevant to the TMLR community.

**Claims And Evidence:**

The paper claims that the local learning rate decay can improve over fedavg (the version with constant client-side step size) and maml adapted to FL (which is an extreme case of only using the late-local-iterate gradient), and empirically validates the proposed method across several FL benchmarks. It also provides convergence guarantees over the proposed algorithm. However, the claimed contributions (along with experiments) have been explored by several previous works (please see 'comment' for details).

**Resubmission Of Major Revision:**

The authors may consider submitting a major revision at a later time.